# MicroRNA exporter HuR clears the internalized pathogens by promoting pro-inflammatory response in infected macrophages

Avijit Goswami[1,†], Kamalika Mukherjee[1,†], Anup Mazumder[1,†,‡], Satarupa Ganguly[1,†], Ishita Mukherjee[2] , Saikat Chakrabarti[2] , Syamal Roy[3], Shyam Sundar[4], Krishnananda Chattopadhyay[2] & Suvendra N Bhattacharyya[1,*]

## Abstract

HuR is a miRNA derepressor protein that can act as miRNA sponge for specific miRNAs to negate their action on target mRNAs. Here we have identified how HuR, by inducing extracellular vesicles-mediated export of miRNAs, ensures robust derepression of miRNA-repressed cytokines essential for strong pro-inflammatory response in activated mammalian macrophages. *Leishmania donovani*, the causative agent of visceral leishmaniasis, on the contrary alters immune response of the host macrophage by a variety of complex mechanisms to promote anti-inflammatory response essential for the survival of the parasite. We have found that during *Leishmania* infection, the pathogen targets HuR to promote onset of anti-inflammatory response in mammalian macrophages. In infected macrophages, *Leishmania* also upregulate protein phosphatase 2A that acts on Ago2 protein to keep it in dephosphory-lated and miRNA-associated form. This causes robust repression of the miRNA-targeted pro-inflammatory cytokines to establish an anti-inflammatory response in infected macrophages. HuR has an inhibitory effect on protein phosphatase 2A expression, and mathematical modelling of macrophage activation process supports antagonistic miRNA-modulatory roles of HuR and protein phosphatase 2A which mutually balances immune response in macrophage by targeting miRNA function. Supporting this model, ectopic expression of the protein HuR and simultaneous inhibition of protein phosphatase 2A induce strong pro-inflammatory response in the host macrophage to prevent the virulent antimonial drug-sensitive or drug-resistant form of *L. donovani* infection. Thus, HuR can act as a balancing factor of immune responses to curtail the macrophage infection process by the protozoan parasite.

**Keywords** Ago2 dephosphorylation; drug-resistant *Leishmania*; host–parasite interaction; miRNA export; protein phosphatase 2A

**Subject Categories** Immunology; Microbiology, Virology & Host Pathogen Interaction

## Introduction

Macrophages act as the first line of defence against the invading microbes in mammalian hosts which engulf the invading pathogens and kill them (Mogensen, 2009). However, the macrophages may fall prey to certain pathogens that inactivate the arsenals of the host macrophage through variety of complex mechanisms (Aderem & Underhill, 1999). The protozoan parasite *Leishmania donovani* (*Ld*), the causative agent of visceral leishmaniasis in human (Ready, 2014), can live and replicate within the host macrophages and thus can escape from the host immune system (Liu & Uzonna, 2012). Macrophage that otherwise gets activated by pathogen-derived molecules remains immunologically dormant when it is invaded by the pathogen *Leishmania* (Olivier *et al*, 2005). The parasite gets into the macrophage through host phagocytic activity and lives within the parasitophorous vacuoles, a special class of endosomal structures that get matured but do not get fused to lysosomes in infected cells through an unique mechanism induced by the internalized pathogen to prevent its own killing by host lysosomal machineries (Courret *et al*, 2002).

The inactivation of the macrophage defence machineries against invading pathogens is achieved through a selective action of pathogen-derived molecules on the host immune system (Contreras *et al*, 2010). *Leishmania* not only impairs the acquired immunity of the host by preventing processing of the pathogen-derived antigens and its presentation by infected macrophage or dendritic cells on

1  RNA Biology Research Laboratory, Molecular Genetics Division, CSIR-Indian Institute of Chemical Biology, Kolkata, India
2  Structural Biology and Bio-informatics Division, CSIR-Indian Institute of Chemical Biology, Kolkata, India
3  National Institute of Pharmaceutical Educations and Research, Kolkata, India
4  Department of Medicine, Banaras Hindu University, Varanasi, India
   *Corresponding author. Tel: +91 33 24995783; E-mails: suvendra@iicb.res.in; sb@csiriicb.in
   †These authors contributed equally to this work
   ‡Present address: University of California Los Angeles, Los Angeles, CA, USA

their surface as part of MHC complex for antibody production (Podi-novskaia & Descoteaux, 2015), but it also ensures reduction in production of nitric oxide and reactive oxygen species in invaded cells to stop killing of the internalized pathogens (Kumar *et al*, 2018). By inactivation of p38/MAPK-driven NF-κB signalling, the transcription of the pro-inflammatory cytokine encoding mRNAs is compromised while production of anti-inflammatory cytokines like IL-10 get enhanced in *Ld* invaded macrophages (Halle *et al*, 2009; Kozicky & Sly, 2015; Kumar *et al*, 2018).

The mechanism of balanced pro- vs. anti-inflammatory cytokine production is a complex and robust process that requires intricate action and crosstalk between cellular signalling machineries and is primarily controlled by several kinases and phosphatases that regulate activation–deactivation states of cellular signalling components linked with regulation of immune response in mammalian macrophages (Kozicky & Sly, 2015; Lloberas *et al*, 2016). *Leishmania* is known to control several of these kinases and phosphatases that are involved in determining the balanced expression of both pro- and anti-inflammatory cytokines (Soulat & Bogdan, 2017).

miRNAs are tiny gene regulatory RNAs that regulate gene expression reversibly by inducing translational suppression and storage or degradation of the repressed messages (Bartel, 2018) in a contextual and candidate specific manner. The action of the miRNAs can get reversed on their targets (Bhattacharyya *et al*, 2006). Derepression of miRNA activity for target cytokine mRNAs is achieved by phosphorylation and miRNA uncoupling of the Ago2 protein in bacterial membrane lipopolysaccharide (LPS)-stimulated macrophage cells (Mazumder *et al*, 2013). This ensures a robust expression of pro-inflammatory cytokines in macrophages exposed to bacterial LPS. Interestingly, the reduction of cytokine mRNA levels after prolonged exposure of macrophage to LPS has also been noted (Mazumder *et al*, 2013) and it has been anticipated as the primary mechanism to check out excess cytokine production in activated macrophage when re-repression of the miRNA-target cytokine mRNAs has been observed. On the contrary, during infection, *Leishmania* is known to upregulate the binding of Ago2 with miRNAs (Chakrabarty & Bhattacharyya, 2017). However, there are other ways to modulate miRNP activity and levels that animal cells adopt under changed context (Patranabis & Bhattacharyya, 2016). Human ELAVL1 protein HuR is known for its anti-miRNA function. The protein, in stressed human hepatocytes, is known to act as a derepressor of miRNA function, where by binding the 3′UTR of common target messages, HuR replaces the bound miRNPs from target mRNAs and ensures uncoupling of miRNAs from the replaced miRNPs. This process is very much determined by miRNAs identity and its binding with HuR that causes accelerated extracellular export of corresponding miRNAs from human hepatocytes under stress (Mukherjee *et al*, 2016). Thus being able to induce derepression of miRNP-mediated target gene repression, HuR should supposedly favour the pro-inflammatory response in mammalian macrophage cells by accelerating expression of miRNA-targeted mRNAs that also have AU-rich elements and thus can be protected by HuR against degradation (Meisner & Filipowicz, 2011).

In this manuscript, we have identified that *Ld* has opposite effects on protein phosphatase 2A (PP2A) and HuR, and thus can eventually determine miRNA-controlled cytokine expression in mammalian macrophages. We have identified PP2A responsible for miRNP recycling in LPS-stimulated macrophage. It ensures

dampening of the pro-inflammatory cytokine production in prolonged LPS-exposed macrophages by promoting re-loading of miRNAs with Ago2 and favours repression of excess cytokine mRNAs in activated cells. PP2A favours anti-inflammatory response in *Ld*-infected macrophages and thereby essential for pathogen survival. On the contrary, HuR, that promotes the inactivation of existing miRNPs and derepresses their target genes, gets inactivated in *Ld*-infected macrophages. Restoration of HuR level, by ectopic expression of the protein in infected cells, reverses the action of the pathogen on cytokine production and clears the parasite from infected host cells or animal tissues. The antimony-resistant *Leishmania donovani* ($Ld^R$), known to induce strong anti-inflammatory response in the host, causes high PP2A expression. Thus, the infection with $Ld^R$ could not be reversed by restoration of HuR level alone but through simultaneous inhibition of PP2A along with ectopic expression of HuR to negate the strong anti-inflammatory effect that the drug-resistant pathogen induces in invaded macrophages by targeting both PP2A and HuR. Finally, taken from the leads of the experimental data presented in this manuscript, the mathematical model of the macrophage activation process suggests a biphasic action of HuR and PP2A that reciprocally ensures a robust control of cytokine expression in mammalian macrophages.

# Results

### PP2A is necessary for Ago2 dephosphorylation

Lipopolysaccharide is an immunostimulatory molecule, derived from the outer membrane of Gram-negative bacteria that acts via TLR4 receptor to activate p38/MAPK pathway in macrophage cells (Bode *et al*, 2012). It augments the NF/kB signalling pathway to induce expression of pro-inflammatory cytokines (Chen *et al*, 2006). The transcriptional surge of new cytokine mRNAs is accompanied by enhanced translation and derepression of miRNP machineries during the early phase of macrophage activation by LPS when cellular miRNPs get deactivated due to phosphorylation of Ago2 and consequent unbinding of associated miRNAs. The derepression of miRNPs thus ensures the surge of pro-inflammatory cytokine production immediately after LPS stimulation (Mazumder *et al*, 2013).

However, on prolonged exposure of LPS, we documented restoration of miRNA activity, despite no major change in total cellular content of miRNA let-7a throughout the early to late stimulatory phases in LPS-treated macrophages (Fig 1A–D). With re-establishment of miRNA activity during long hours/late phase of LPS stimulation, levels of miRNA-regulated cytokine mRNAs IL-6 and TNF-α were also dropped to a much reduced level compared to high cytokine mRNAs observed in early phase of activation due to loss of miRNA activity there [Fig 1C (Mazumder *et al*, 2013)]. Cellular levels of miRNA let-7a, a miRNA known to control IL-6 mRNA, did not alter with LPS exposure but its association with Ago2 was relieved initially and then reversed over the course of LPS exposure (Fig 1D).

Reactivation of miRNPs can account for increased miRNA activity, and repression of target IL-6 mRNA was observed in prolonged LPS-treated cells. However, it should be preceded by dephosphorylation of Ago2 necessary for re-loading of Ago2 with miRNAs.

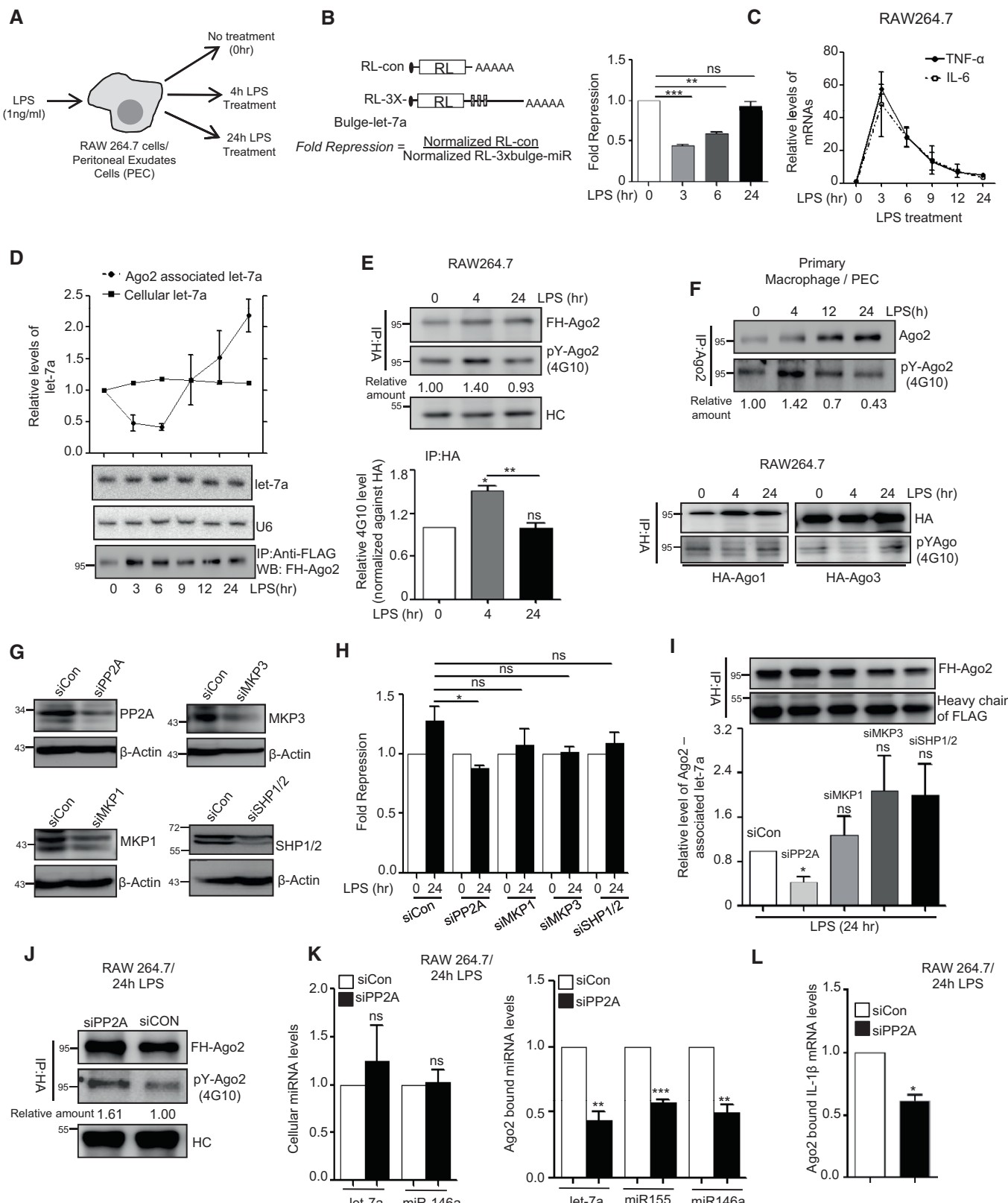

**Figure 1.**

◀

**Figure 1.  PP2A-dependent Ago2 dephosphorylation during prolonged LPS stimulation of mammalian macrophage cells.**

A–C   Reversal of miRNP activity and its restoration during the LPS treatment of RAW 264.7 cells. Schematic representation of the experiments where RAW 264.7 or PEC cells were treated with 1 ng/ml of LPS for different time points before the RNA and protein content were analysed (A). Relative fold repression for let-7a reporter in RAW 264.7 cells during LPS treatment (mean ± s.e.m., n = 3) (B). The schematic representation of the constructs used for luciferase assays is shown. Relative levels of pro-inflammatory cytokines TNF-α and IL-6 mRNA, a target of let-7a miRNA, have been plotted against time of LPS stimulation, and 18S rRNA was used for normalization (mean ± s.e.m., n = 2 (IL-6), n = 3 (TNF-α)) (C).

D–F   Time-dependent Ago2 phosphorylation and its miRNA binding in LPS-treated macrophage cells. RAW 264.7 cells were transfected with FH-Ago2 expression plasmid and treated with LPS (1 ng/ml) for different time points. Ago2 was pulled down using anti-FLAG beads, and the levels of Ago2-associated let-7a miRNA were measured by qRT–PCR. Quantity of immunoprecipitated Ago2 was detected in Western blot analysis with anti-HA antibody and used for normalization of amounts of miRNA detected by qRT–PCR. Levels of cellular let-7a were measured by Northern blot, and U6 RNA was used as loading control. Relative levels of cellular and Ago2-bound let-7a were plotted (mean ± s.e.m., n = 3) (D). Phosphorylated Ago2 (pY-Ago2) level was measured using 4G10 antibody specific for phosphorylated Tyrosine. Phosphorylated Ago2 (pY-Ago2) band intensities were normalized against total Ago2 detected with HA-specific antibody that was also used for Ago2 pull down (E, *upper panel*). Relative 4G10 intensities were plotted from three independent experiments (mean ± s.e.m.) (E, *lower panel*). Primary cells (PEC) were isolated from BALB/c mice and endogenous Ago2 was immunoprecipitated with anti-Ago2 (eIF2C2) antibody and phosphotyrosine level of Ago2 was measured using 4G10 antibody. Relative intensities of phospho-Ago2 against total amount of immunoprecipitated Ago2 were quantified and mentioned below the lanes (mean ± s.e.m., n = 3) (F, *upper panel*). Similar experiments were done with FH-Ago1 and Ago3, and levels of phosphotyrosine in immunoprecipitated FH-Ago1 and FH-Ago3 were measured in RAW 264.7 cells (F, *lower panel*).

G–I   Effect of knock-down of different phosphatases on miRNA-mediated repression and miRNA association of Ago2 in 24-h LPS-treated RAW 264.7 cells. Knock-down of PP2A, MKP1, MKP3 or SHP1/2 was checked by Western blot analysis done with lysates of cells treated with respective siRNAs. β-Actin was used as loading control (G). Luciferase-based let-7a miRNA reporter assay was done in LPS-treated cells downregulated for specific phosphatases. Levels at 0-h time point were taken as units (mean ± s.e.m., n = 4) (H). Ago2-associated let-7a was estimated by qRT–PCR in LPS-treated cells depleted for specific phosphatases. let-7a miRNA level was normalized against respective FH-Ago2 bands (mean ± s.e.m., n = 3) (I).

J–L   Effect of PP2A knock-down on Ago2 phosphorylation and its miRNA association. Phospho Ago2 levels were measured in siCon- and siPP2A-transfected cells upon 24 h of LPS treatment using phosphotyrosine specific 4G10 antibody. HA-Ago2 was detected with anti-HA antibody (J). Cellular miRNA levels (K, left panel mean ± s.e.m., n = 3) and Ago2-associated miRNAs content in cells treated with siCon or siPP2A were measured using qRT–PCR and plotted (K, right panel, mean ± s.e.m., n = 3). Ago2-associated IL-1β mRNA level was estimated in siCon- and siPP2A-treated cells which were stimulated with LPS for 24 h (mean ± s.e.m., n = 3) (L). Immunoprecipitated Ago2 content was used for normalization for both miRNA and mRNA levels in above-mentioned experiments.

Data information: In all experimental data, ns: non-significant and *, ** and *** represent *P*-value of < 0.05, < 0.01 and < 0.001, respectively, calculated by using Student's *t*-test. HC: heavy chain of the antibody used for immunoprecipitation. For statistical analysis, all experiments were done minimum three times. Exact *P*-values against each experimental set are presented in the Appendix Table S2. Positions of molecular weight markers are shown in the Western blots used in different panels. Source data are available online for this figure.

During this long exposure of LPS, we have documented reversal of Ago2 phosphorylation that had increased during early phase of LPS treatment of RAW 264.7 cells (Fig 1E). Interestingly, Ago1 and Ago3 showed an opposite effect upon LPS activation as they showed restoration of phosphorylation after 24 h of LPS exposure while a decrease in phosphorylation was noted at 4 h of LPS exposure (Fig 1F, *lower panel*). Cellular level of Ago2, like that of miRNA, was not altered during early or late activation phases (Figs 1E and EV1A). Similar results were also observed in primary macrophage cells (Fig 1F, *upper panel*).

To identify the potential candidate phosphatase that acts on phosphorylated Ago2 to reactivate it in prolonged LPS-treated cells, we first checked the expression of key phosphatases in LPS-treated cells that are known to get upregulated in LPS-activated macrophages (Kozicky & Sly, 2015). We observed an increased expression of protein phosphatase 2A (PP2A) and mitogen-activated protein kinase phosphatase 1 (MKP1) in activated macrophage (Fig EV1A and B). To specify the phosphatase that is responsible for Ago2 dephosphorylation, we targeted them individually with specific sets of siRNAs. We have noted PP2A as the effective phosphatase that acted on phosphorylated Ago2 in LPS-treated macrophage, and as observed, depletion of PP2A showed defective recovery of let-7a miRNP activity and its Ago2 association during prolonged LPS activation. Effect of siRNA-mediated depletion of two other phosphatases *MKP3* and *SHP1/2* showed no inhibitory effect of miRNA-activity recovery during LPS treatment (Fig 1G–I). siRNA-mediated PP2A knock-down was also effective to prevent the re-binding of the miRNAs to Ago2 but without a change in cellular miRNA content upon prolonged activation by LPS. As expected, siPP2A treatment was also associated with increased Ago2 phosphorylation

and decreased target RNA binding of Ago2 (Fig 1J–L). Interestingly, similar increase in PP2A levels were noted in RAW 264.7 cells treated with PMA or in primary macrophage treated with LPS. Increased levels of PP2A mRNAs were also noted in RAW 264.7 cells upon LPS treatment (Fig EV1C–E).

**Phosphorylation–dephosphorylation cycle of Ago2 controls miRNP recycling and *de novo* miRNP formation**

Like siPP2A treatment, application of okadaic acid (OA), a chemical inhibitor of PP2A (Fernandez *et al*, 2002), inhibited the Ago2 dephosphorylation and restoration of miRNA activity in prolonged LPS-treated macrophages (Fig 2A–C). Incubation of the PP2A immunoprecipitated materials with FLAG-HA-tagged Ago2 isolated from a non-immune human cell HEK293, the phosphorylated Tyrosine at 529 of Ago2 got dephosphorylated in the *in vitro* phosphatase assay. Interestingly, unlike the wild type, the Ago2-Y529F mutant that could not get phosphorylated at 529 position showed no change in its phosphorylation status upon incubation with PP2A immunoprecipitated materials. These data suggest that PP2A specifically dephosphorylates Ago2 at Y529 position and thus should be effective in controlling miRNA binding of Ago2 as phosphorylation at Y529 position of Ago2 is known to cause miRNA unbinding [Fig 2D (Rudel & Meister, 2008)]. Use of OA, the inhibitor of PP2A, increased the phosphorylated form of Ago2. OA should therefore inhibit *de novo* miRNP formation in prolonged LPS-treated cells by reducing the cellular free Ago2 pool ready for miRNA loading. As expected, inducible expression of miR-122, that got expressed in the presence of doxycycline in cells otherwise not expressing the liver-specific miR-122 but harbouring a doxycycline inducible miR-122

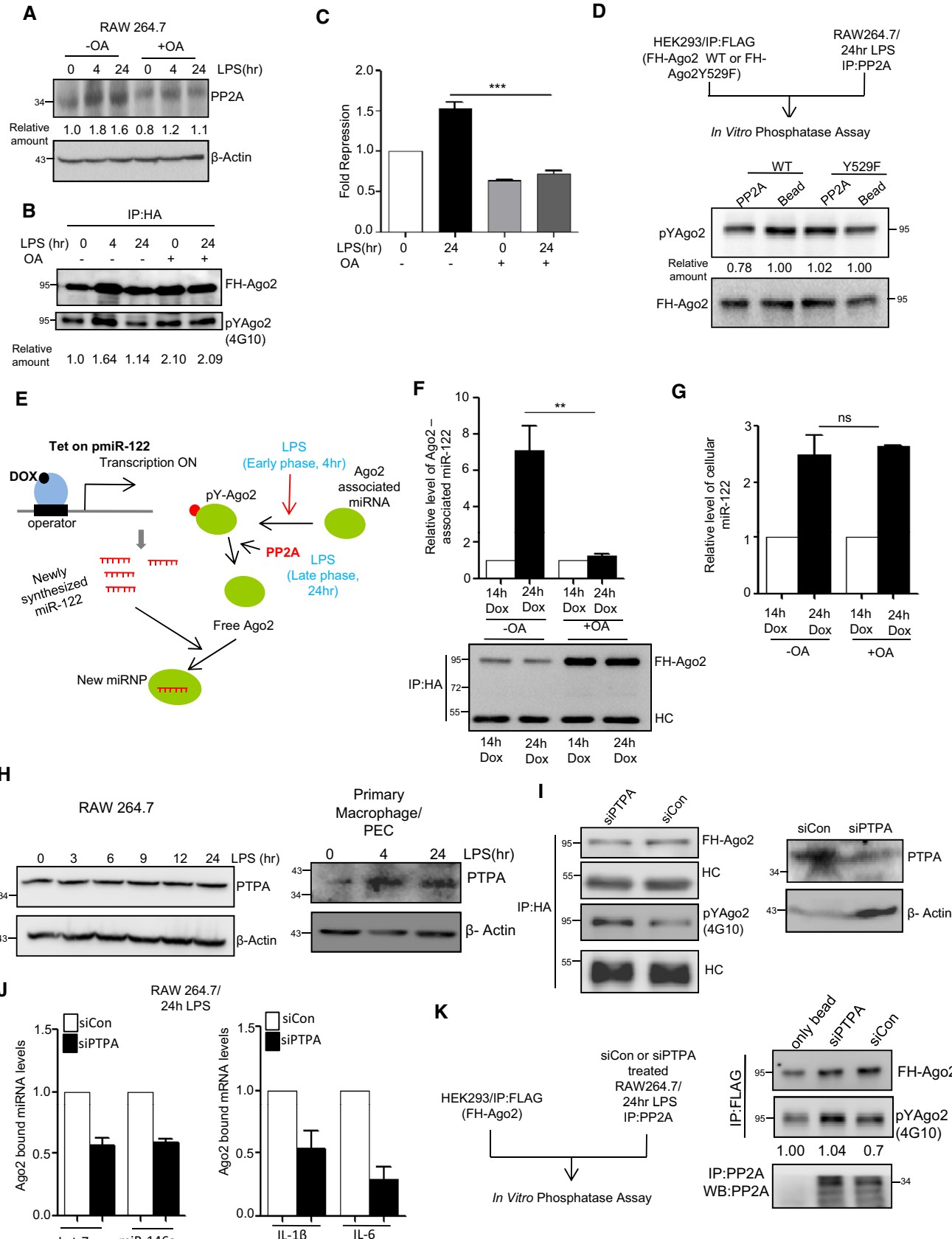

**Figure 2.**

**Figure 2.  Phosphorylation–dephosphorylation cycle of Ago2 controls miRNP recycling and *de novo* miRNP formation.**

A–C   Effect of okadaic acid (OA; 100 nM), the PP2A inhibitor, on Ago2 phosphorylation and miRNA activity. Expression of PP2A was detected in Western blot done for cell extract from control and OA-treated RAW 264.7 cells after LPS stimulation (A). The amount of phosphorylated Ago2 was measured in OA-treated cells before and after LPS stimulation. The amount of Tyr phosphorylated Ago2 was measured by densitometric quantification of Western blot data upon normalization against immunoprecipitated Ago2 amount (B). Changes in fold repression for a let-7a reporter upon LPS exposure in control and OA-treated cells (mean ± s.e.m., $n$ = 3) (C).

D   Schematic representation of the *in vitro* phosphatase assay (*upper panel*). PP2A was immunoprecipitated from 24-h LPS-treated RAW 264.7 cells and was incubated *in vitro* with wild type or phosphorylation defective FH-Ago2 mutant (Ago2Y529F) isolated from HEK293 cells transfected with respective expression constructs. Phosphorylated Ago2 (pYAgo2) levels were measured by Western blot analysis using anti-phosphotyrosine-specific 4G10 antibody. Relative intensities were quantified by densitometric analysis and mentioned below the respective panels. FH-Ago2 was detected by anti-HA antibody (*Lower panel*).

E–G   Defective *de novo* miRNP formation in cells pre-treated with PP2A inhibitor. A schematic representation of the assay done to check the *de novo* miRNP formation after doxycycline (DOX)-induced expression of miR-122, a liver-specific miRNA, in RAW 264.7 cells during LPS stimulation (E). Ago2-associated (F) and total (G) miR-122 levels were measured by qRT–PCR and normalized against immunoprecipitated FH-Ago2 content and U6 RNA, respectively. Values obtained upon 14 h of DOX treatment were considered as unit, and relative values obtained upon 24 h of DOX treatment are plotted. All samples were treated for 24 h with LPS (mean ± s.e.m., $n$ = 4).

H–K   Phosphotyrosyl phosphatase activator (PTPA) is an essential factor for PP2A-mediated dephosphorylation of Ago2. Expression of PTPA after LPS stimulation in RAW 264.7 cells as well as in PEC isolated from BALB/c mice. β-Actin was used as loading control (H). Effect of PTPA depletion on Ago2 phosphorylation level in RAW 264.7 cells (I). Phosphorylated Ago2 level was measured by Western blot in HA-immunoprecipitated materials isolated from either siCon- or siPTPA-transfected cells expressing FH-Ago2 upon 24 h of LPS treatment using a phosphotyrosine-specific 4G10 antibody (I, *left panel*). The PTPA was detected with anti-PTPA antibody, and β-actin was used as loading control (I, *right panel*). Effect of PTPA depletion on Ago2-associated miRNAs and mRNAs. Ago2-associated miRNA (J, *left panel*) (mean ± s.e.m., $n$ = 3) and mRNA levels (J, *right panel*) (mean ± s.e.m., $n$ = 2) were also quantified using qRT–PCR. Amount of FH-Ago2 in immunoprecipitated materials (shown in panel I) were used for normalization. Schematic depiction of *in vitro* phosphatase assay (K, *left panel*). PP2A was immunoprecipitated from siCon- or siPTPA-treated RAW 264.7 cells treated with LPS for 24 h and was incubated *in vitro* with FH-Ago2 isolated from FH-Ago2 stable HEK293 cells and upon the assay reaction, the phospho-Ago2 level was measured by Western blot analysis using phosphotyrosine-specific 4G10 antibody (K, *right panel*).

Data information: Relative quantification was done by densitometry. HC: heavy chain of respective antibodies used for immunoprecipitation. In all experimental data, ns: non-significant and ** and *** represent *P*-value of < 0.01 and < 0.001, respectively, estimated by using Student's *t*-test. Exact *P*-values against each experimental set are presented in the Appendix Table S2. Positions of molecular weight markers are marked and shown in the Western blots used in different panels.
Source data are available online for this figure.

expression system, failed to get incorporated to Ago2 in OA-treated cells with time. Thus, the levels of *de novo* miRNPs got reduced when PP2A was inhibited (Fig 2E–G). These results support the importance of phosphorylation–dephosphorylation cycle of existing miRNPs for *de novo* miRNP formation in mammalian cells that happens during late phase of LPS exposure of macrophages. Thus, phosphorylated Ago2 that undergoes PP2A-mediated dephosphorylation acts as substrate for subsequent *de novo* miRNA loading during late phase of LPS treatment of mammalian macrophages (Fig 2E–G).

**PP2A requires Phosphotyrosyl phosphatase activator (PTPA) for dephosphorylation of Ago2**

The phosphorylation of Ago2 at Tyrosine 529 was found to be downregulated in the presence of PP2A, and Tyr phosphorylation of Ago2 showed an inverse correlation with PP2A levels in macrophage cells. Interestingly, PP2A is a serine-specific phosphatase and should act on proteins with a phosphorylated serine. However, PP2A is also known to act on phosphotyrosine in the presence of an adaptor protein PTPA that could render PP2A to dephosphorylate Tyrosine (Janssens *et al*, 1998). Unlike PP2A, we detected no increase in PTPA expression in macrophage during LPS treatment (Fig 2H). However, with depletion of PTPA we detected reduced Ago2 dephosphorylation in 24 h LPS-treated RAW 264.7 cells (Fig 2I). We also checked the status of Ago2 binding of miRNA and target mRNAs in PTPA-depleted cells and have found reduced miRNA and target RNA association of Ago2 upon LPS treatment in cells depleted for PTPA (Fig 2J). We also checked the *in vitro* phosphatase activity of immune-isolated PP2A from siControl- or siPTPA-treated cells and had found reduced phosphatase activity of PP2A obtained from PTPA-depleted cells on Tyrosine phosphorylated Ago2 during the *in vitro* phosphatase assays (Fig 2K).

**Protein phosphatase 2A upregulation in *Leishmania*-infected macrophages reduces phosphorylated Ago2 and ensures robust repression of pro-inflammatory cytokines**

The pathogen *Ld* infects the host macrophage and evades the host immune response by inhibiting pro-inflammatory cytokine production. This can be easily achieved by sustained miRNP-mediated gene repression of target pro-inflammatory cytokine encoding mRNAs in infected cells. Retention of miRNPs with targets can be achieved by keeping the phosphorylation of Ago2 at minimal level. Ago2 phosphorylation happens primarily during LPS or PMA-mediated activation of macrophages (Mazumder *et al*, 2013). As expected, we noted a decreased pro-inflammatory cytokine production and increased anti-inflammatory IL-10 production both in RAW 264.7 and murine primary macrophages (PEC) upon *Ld* infection (Fig 3A–C). It was accompanied by a decrease in the level of Ago2 phosphorylation and increased Ago2-miRNA binding happening in infected cells (Fig 3D and E). Interestingly, the PP2A level was also increased upon *Ld* infection both in mouse primary macrophage PEC and RAW 264.7 cells (Fig 3B).

Working on the mechanism of *Ld*-induced expression of PP2A in macrophages, we did not detect any increase of PP2A mRNA or protein in RAW 264.7 cells upon treatment with heat killed *Ld* (Fig EV2A and B). Upon interaction with macrophages, *Ld* may elicit the response on PP2A expression either due to phagocytic internalization of the pathogen or due to interaction of surface ligands of *Ld* with receptors present on macrophage membrane.

We used inert latex beads that are internalized through phagocytosis in a time-dependent manner in RAW 264.7 cells (Akilbekova *et al*, 2015). However, no increase in PP2A expression was documented in cells interacting with latex beads rather a decrease in PP2A protein and mRNA level was evident (Fig EV2C–E). Lipophosphoglycan or LPG is a major surface molecule of *Leishmania* in the

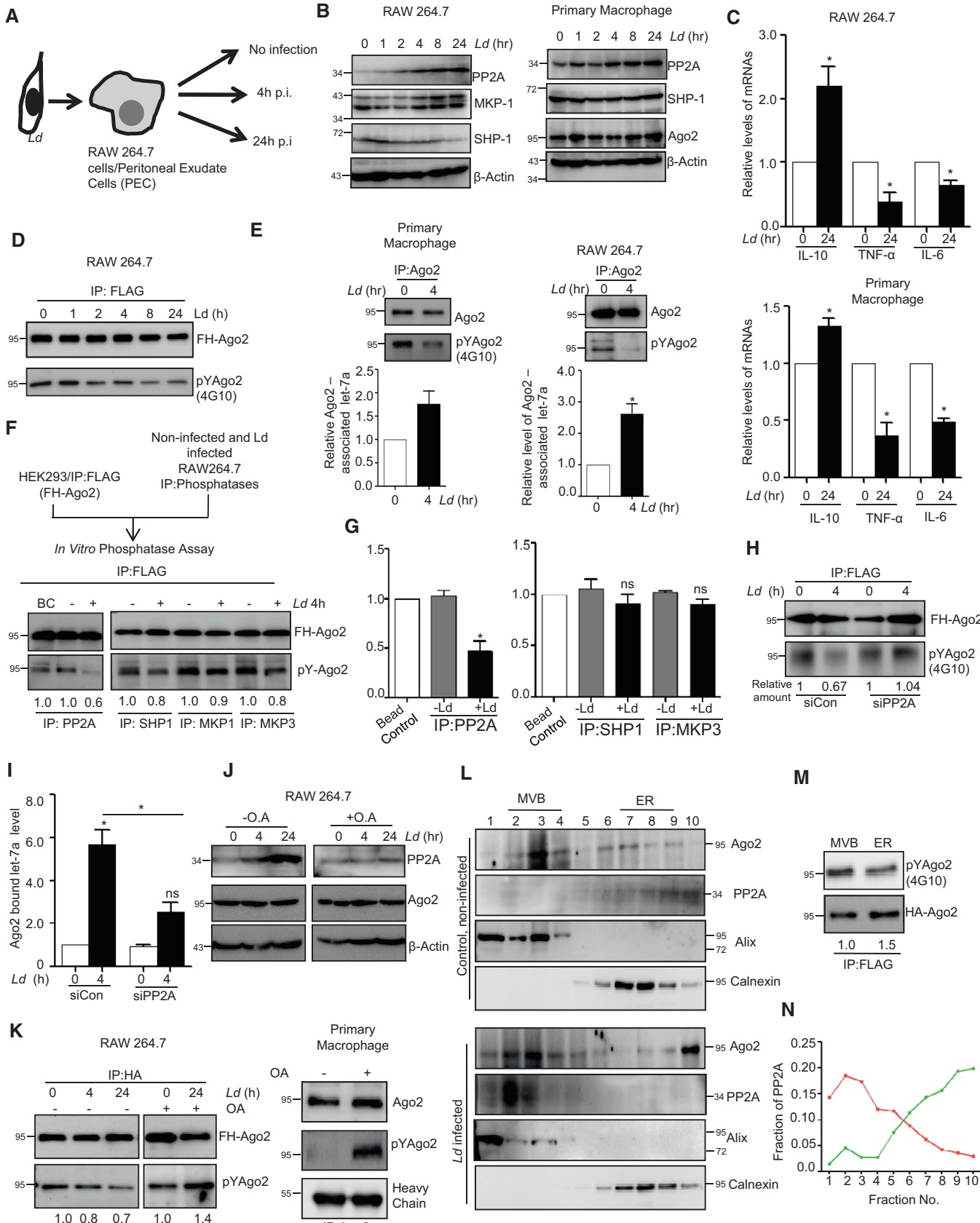

Figure 3.

**Figure 3.** *Leishmania donovani* upregulates PP2A to dephosphorylate Ago2 in infected macrophage cells.

A–C Expression of different cytokines and phosphatases in RAW 264.7 cells and PEC infected with *Leishmania donovani* (*Ld*) parasites. A scheme of the experiments is shown in panel A. Cells at different time points of infection were harvested, and levels of various phosphatases were checked by Western blot analysis using antibodies specific for different phosphatases in both RAW 264.7 (B, left panel) and mouse peritoneal macrophage cells PEC (B, *right panel*). Cytokine mRNA levels like IL-10, TNF-α and IL-6 were quantified after 24 h of *Ld* infection in RAW 264.7 (C, upper panel mean ± s.e.m., *n* = 3) and mouse peritoneal macrophage PEC (mean ± s.e.m., *n* = 3) (C, Lower panel).

D, E Effect of *Ld* infection on Ago2 phosphorylation and its miRNA association. RAW 264.7 cells were transfected with FH-Ago2 expression construct, and infection was done for various time points followed by FH-Ago2 pull down using anti-FLAG beads. Phosphorylated Ago2 levels were checked during the course of *Ld* infection (D). Phosphorylation levels of Ago2 were measured by doing endogenous Ago2 pull down using Ago2-specific antibody after 4 h of infection and quantified along with Ago2-associated let-7a level measurement in primary macrophages (mean ± s.e.m., *n* = 2) (E, *left panel*) and in RAW 264.7 cells (mean ± s.e.m., *n* = 3) (E, *right panel*).

F, G Schematic representation of *in vitro* phosphatase assay (F, *upper panel*). PP2A, MKP1, MKP3 or SHP1 was immunoprecipitated individually from naïve or *Leishmania*-infected macrophage cells and was incubated *in vitro* with FH-Ago2 isolated from FH-Ago2 stable HEK293 cells. Phosphorylated Ago2 level was detected by Western blot analysis using phosphotyrosine-specific 4G10 antibody (F, lower panel) and measured by densitometry. Relative intensity of 4G10 specific band against immunoprecipitated Ago2 was plotted (mean ± s.e.m., *n* = 3) (G). BC: bead control.

H, I Effect of PP2A knock-down on Ago2 phosphorylation and miRNA-Ago2 binding in RAW 264.7 cells infected with *Ld*. Cells were co-transfected with siRNA and FH-Ago2 expression plasmid. Phosphorylated Ago2 levels were detected after 4 h of infection in siCon- or siPP2A-transfected cells (H). Ago2-associated let-7a levels were also estimated and plotted (mean ± s.e.m., *n* = 4) (I).

J, K Effect of OA on Ago2 phosphorylation in *Ld*-infected RAW 264.7 cells and mouse PEC. Cellular levels of PP2A and Ago2 at different time points of *Ld* infection with or without OA treatment (100 nM) were detected. Cells were pre-treated with OA (100 nM) for 2 h before infection (J). β-Actin was used as loading control. Phospho-Ago2 level was measured in OA pre-treated; *Ld*-infected RAW 264.7 cells expressing FH-Ago2 and in PEC after 24 h of infection with or without OA pre-treatment. Phosphorylated Ago2 levels were also checked (K).

L–N Subcellular compartmentalization of PP2A in *Ld*-infected RAW 264.7 cells. RAW 264.7 cells were infected with *Ld* for 6 h, and cell extract was analysed on a OptiPrep^R density gradient to separate subcellular organelles and structures. Subcellular localization of Ago2 and PP2A in individual fractions was detected by Western blots. Alix was used at MVB marker, while Calnexin was used as ER marker (L). OptiPrep^R fractions, positive for MVB (fraction number 2,3,4) and ER (fraction number 7,8,9) markers, were pooled separately, and Ago2 was immunoprecipitated using anti-FLAG beads. Phosphorylated Ago2 levels were quantified in Western blot done with phosphotyrosine-specific 4G10 antibody (M). PP2A percentage intensity plot for each fraction was done for control (Green line) and infected (Red line) samples (N).

Data information: In all experimental data, ns: non-significant and * represents *P*-value of < 0.05 estimated by using Student's *t*-test. HC: heavy chain of respective antibody used for immunoprecipitation. Exact *P*-values against each experimental set are presented in the Appendix Table S2. Positions of molecular weight markers are marked and shown in the Western blots used in different panels.

Source data are available online for this figure.

promastigote stage and has been reported to interact with several Toll-like receptors (TLRs) (Becker *et al*, 2003; Rojas-Bernabe *et al*, 2014; Chaparro *et al*, 2019). PP2A upregulation during *Ld* infection has been previously reported (Kar *et al*, 2010); however, the exact mechanism is unknown. We speculated that upregulation of PP2A could be mediated by TLR pathway upon interaction with LPG. To check whether LPG mediates PP2A upregulation, RAW 264.7 cells were treated with LPG isolated from *Ld*. Interestingly, LPG was found to upregulate PP2A mRNA level in a concentration-dependent manner (Fig EV2F–I). However, treatment of cells with anti-TLR4 antibody to block the TLR4 activation by LPS has been effective to reduce PP2A expression in RAW 264.7 cells and production of TNF-α mRNA there (Fig EV2F and G). Using the same anti-TLR4 antibody, we could reduce the expression of PP2A induced by LPG derived from *Ld* (Fig EV2H and I).

To confirm that the increased PP2A, present in the infected macrophage, is responsible for decreased Ago2 phosphorylation observed, we performed an *in vitro* phosphatase assay with affinity purified FH-Ago2 isolated from HEK293 cells and treated them with immunoprecipitated phosphatases from *Ld*-infected RAW 264.7 cells individually. Substantial decreases in Ago2 phosphorylation level were only observed upon interaction with PP2A immunoprecipitated materials suggesting PP2A as the major phosphatase present in *Ld*-infected cells that is responsible for decreased Ago2 phosphorylation (Fig 3F and G). The conclusion was substantiated in another set of experiments where cells were transfected with siRNAs against PP2A, and an increase in Ago2 phosphorylation and decrease in Ago2-bound miRNA content were noted in siPP2A-treated cells compared to control siRNA-treated cells upon *Ld*

infection (Fig 3H and I). Similarly, treatment of OA, the chemical inhibitor of PP2A, showed no effect on Ago2 levels but showed an increase in Ago2 phosphorylation in *Ld*-infected macrophage cells (Fig 3J and K).

The compartment where PP2A prevents the pro-inflammatory response via enhancing the miRNPs stability in *Ld*-infected cells was of interest, and we explored that in subsequent experiments. We isolated the different subcellular organelles over an Optiprep^R density gradient that could separate out the organelles on the basis of their densities (Fig 3L). We noted an increased association of PP2A with the endosome enriched fraction in *Ld*-infected cells (Fig 3N). It is important to note that substrate of PP2A, the phosphorylated Ago2, is also known to get concentrated in these fractions as we had analysed them before both in macrophage and neuronal cells [Fig 3L and M (Mazumder *et al*, 2013; Patranabis & Bhattacharyya, 2016)]. Therefore, PP2A may act on its substrate phosphorylated Ago2 in the Optiprep density gradient fractions 2–4 that are positive for MVB marker protein Alix.

## Protein phosphatase 2A-mediated stabilization of miRNPs ensures anti-inflammatory response and *Ld* infection in mammalian macrophages

*Ld* upregulate PP2A to promote anti-inflammatory responses (Kar *et al*, 2010). In support of this notion, we have identified the increased production of IL-10, the hallmark anti-inflammatory cytokine expressed in *Ld*-infected mammalian macrophages. Depletion of PP2A by siRNA or its inhibition by OA resulted in downregulation of IL-10 production in *Ld*-infected cells (Fig 4A). This was accompanied

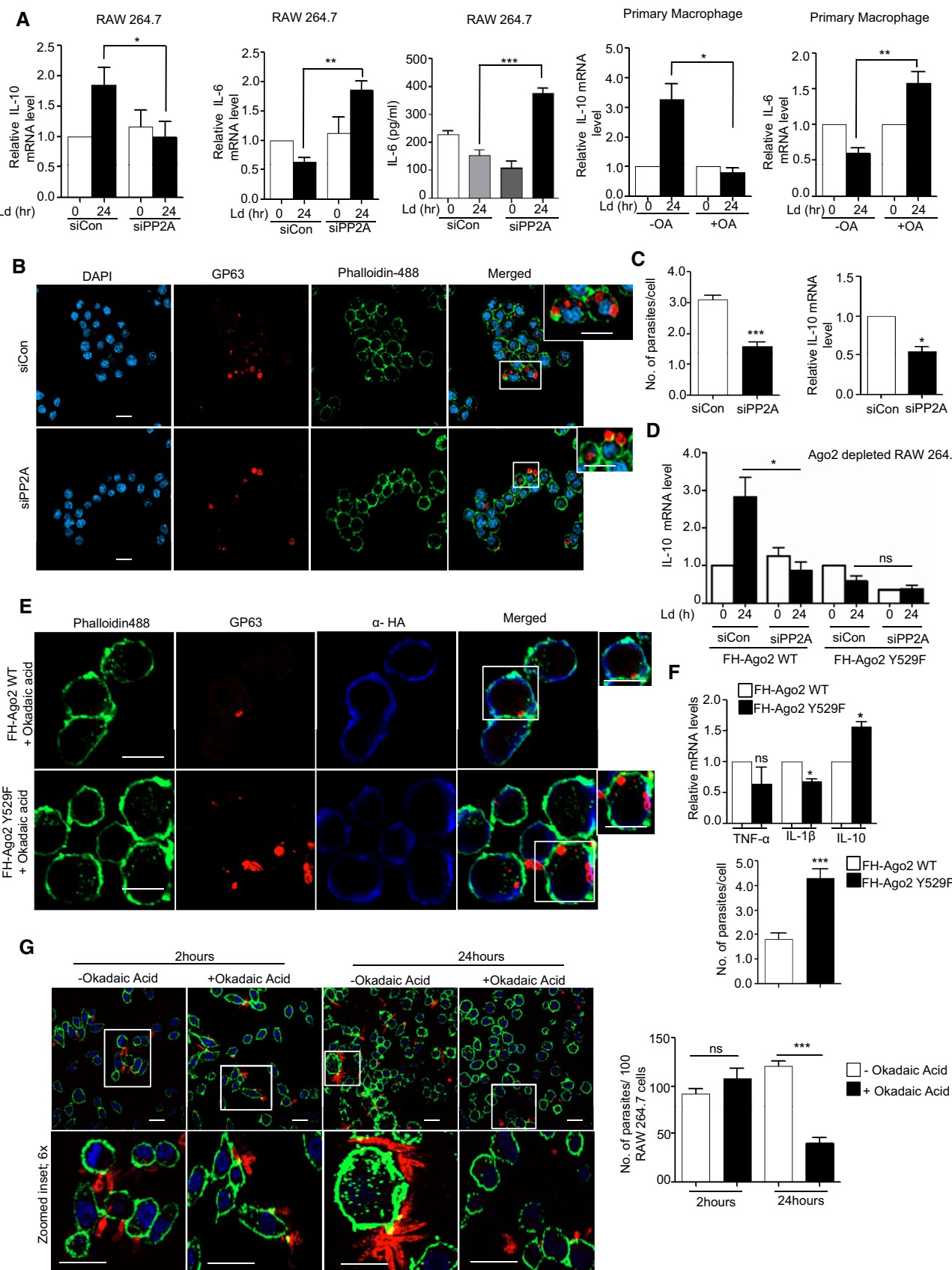

**Figure 4.**

◀

**Figure 4.  PP2A-mediated dephosphorylation of Ago2 is necessary for *Leishmania donovani* infection of macrophage cells.**

A   Effect of PP2A downregulation or inhibition on pro- and anti-inflammatory cytokine levels in *Ld*-infected macrophage. IL-10 levels were measured after *Ld* infection in PP2A knocked-down RAW 264.7 cells (mean ± s.e.m., n = 3) and in PEC (mean ± s.e.m., n = 3). IL-6 was also measured in RAW cells after OA treatment (mean ± s.e.m., n = 5) and also in PEC pre-treated with OA (mean ± s.e.m., n = 3). Values obtained at 0 h (non-infected) or non-OA treated samples were considered as unit in each case. IL-6 protein level (n = 5) in cell supernatant was also measured by ELISA after *Ld* infection in PP2A knock-down RAW 264.7 cells.

B, C   Effect of PP2A knock-down on parasite internalization and cellular IL-10 level. Internalized parasites were imaged in RAW 264.7 cells stained with Phalloidin-Alexa 488 (green) for actin cytoskeleton detection, and *Ld* was detected by indirect immunofluorescence done for parasite specific membrane protein GP63 (red) (B). Number of *Ld* internalized was measured in control and PP2A knock-down cells, and relative numbers of parasites within infected cells were plotted [C, left panel (mean ± s.e.m., n = 20 cells)]. IL-10 mRNA levels were also quantified in those cells by qRT–PCR done for total cellular RNA [C, right panel (mean ± s.e.m., n = 3)]. Scale bar 20 μm. 4× Zoomed insets are shown.

D   Effect of PP2A knock-down on IL-10 level in *Ld*-infected cells expressing phosphorylation defective mutant of Ago2. In RAW 264.7 cells, depleted for endogenous Ago2 (using a specific siRNA against the 3′UTR of Ago2 mRNA), either the wild type or Y529F mutant of FH-Ago2 (without having the 3′UTR of Ago2) was expressed in siCon- or siPP2A-treated RAW 264.7 cells. IL-10 mRNA level was measured subsequent to *Ld* infection. Relative IL-10 levels were plotted. Expression levels in non-infected siCon-treated cells in each case were designated as unit (mean ± s.e.m., n = 3).

E–G   Effect of PP2A inhibitor OA on *Ld* internalization and cytokine production in RAW 264.7 cells. In RAW 264.7 cells transfected with FH-Ago2 WT and Y529F mutant and pre-treated with OA, *Leishmania* internalization was measured microscopically. In images obtained, Ago2 was detected with α-HA (detected at 405 nm) and *Leishmania* was stained for GP63 (detected at 564 nm) (E). Quantitative measurement of GP63-positive structures was done, and quantitative data per 100 infected cells were plotted. Scale bar 10 μm. RNA was isolated from different experimental sets, and TNF-α, IL-1β and IL-10 mRNA levels were estimated (mean ± s.e.m., n = 3) (F). RAW 264.7 cells with and without OA treatment were given *Ld* infection for 2 and 24 h. RAW 264.7 cells were stained with Phalloidin-Alexa 488 (green) for actin cytoskeleton, and *Ld* was stained for parasite specific protein GP63 (red) (G, *left panel*). Number of internalized parasites per 100 macrophages were calculated, and relative values were plotted (G, *right panel*) (mean ± s.e.m., n = 20). Scale bar 20 μm.

Data information: Zoomed parts are been highlighted with white boxes in the microscopic pictures. In all experimental data, ns: non-significant and *, ** and *** represent *P*-value of < 0.05, < 0.01 and < 0.001, respectively, quantified with the help of Student's *t*-test. For statistical analysis, all experiments are done minimum three times. Exact *P*-values against each experimental set are presented in the Appendix Table S2. Positions of molecular weight markers are marked and shown in the Western blots used in different panels.

Source data are available online for this figure.

by increased production of pro-inflammatory cytokine IL-6 both at mRNA and protein level (Fig 4A). Interestingly, internalization of the pathogen was reduced in the presence of siPP2A (Fig 4B and C, *left panel*). It was accompanied by decreased production of anti-inflammatory IL-10 in siPP2A-treated *Ld*-infected macrophages (Fig 4C, *right panel*). Importance of Ago2 dephosphorylation in the infection process was studied further in Ago2-depleted cells by expressing either the wild-type Ago2 or phosphorylation defective Y529F Ago2 mutant co-transfected either with siCon or siPP2A. Interestingly, unlike the FH-Ago2-expressing cells where depletion of PP2A strongly reduces IL-10 levels, there was little effect of PP2A depletion on IL-10 production in FH-Ago2Y529F mutant expressing cells (Fig 4D). This result supports the notion that the anti-inflammatory effect of PP2A induced by *Ld* infection is primarily through the control of phosphorylation–dephosphorylation cycle of Ago2 in infected cells. Therefore, PP2A has little effect on infection induced IL-10 level change in Ago2Y529F-expressing cell, as in those cells, Ago2Y529F is phosphorylation defective and therefore should not be responsive to the presence and absence of PP2A (Fig 4D). The importance of PP2A-mediated Ago2 dephosphorylation in infection process was further supported in other experiments where the expression of the Ago2Y529F mutant, compared to its wild-type counterpart, showed increased production of cytokine IL-10 and better internalization of parasite into the cells pre-treated with OA (Fig 4E). It was accompanied by a decrease in the expression of pro-inflammatory cytokine IL-1β in OA-treated RAW 264.7 cells expressing Ago2Y529F (Fig 4F). In cells expressing wild-type Ago2, OA showed inhibition of *Ld* internalization and IL-10 production as PP2A is inhibited there. These data support the notion that in cells expressing a phosphorylation defective mutant of Ago2, the strong repression of pro-inflammatory cytokines would occur, and thus, they should be more susceptible to infection. The phosphorylation of Ago2 and its uncoupling with miRNA thus serve as an important mechanism of inflammatory response that got reversed during *Ld* infection in macrophage with wild-type Ago2 but

not in cells expressing Ago2Y529F (Fig 4D–G). These data also suggest a late effect of PP2A inhibition on internalized parasites as at early 2-h time no major change in internalized parasite number was detected (Fig 4G).

### HuR-mediated export of miRNA is necessary and sufficient for pro-inflammatory immune response in mammalian macrophages

Balancing of pro-inflammatory responses by anti-inflammatory pathways in infected macrophage may need targeting of pro-inflammatory factors by the pathogen. In this context, miRNA derepressor protein HuR (Bhattacharyya *et al*, 2006) should have been an interesting candidate to explore as miRNA derepression due to miRNA-Ago2 uncoupling has been found to be a prerequisite for pro-inflammatory response in activated macrophage cells (Mazumder *et al*, 2013). Interestingly, there was increased level of HuR, the miRNA uncoupler, in macrophages activated with LPS (Fig 5A and B). It has been reported already that HuR is necessary and sufficient for miRNA export from mammalian hepatic cells under cellular stress (Mukherjee *et al*, 2016). Human ELAVL1 protein HuR has three RNA binding motifs, and it binds and sponges out specific miRNAs (Kundu *et al*, 2012; Poria *et al*, 2016). While HuR binds to mRNAs with AU-rich sequences, it could reversibly bind specific miRNA with AU/G sequences and may facilitate their export via exosomes, a special class of extracellular vesicles (EVs) (Mukherjee *et al*, 2016). As HuR level increases with increase in expression of canonical HuR target mRNAs like TNF-α in activated macrophages (Chen *et al*, 2006), the export of miRNAs is also expected to be accelerated in activated macrophages. HuR was found to be upregulated in those cells, and miRNAs that otherwise may play a repressive action on HuR target messages should be inactivated there. We explored the status of miRNA export from LPS-treated cells and noticed an accelerated miRNA export upon LPS exposure of macrophage cells. The characteristic profile of extracellular vesicles (EVs) isolated

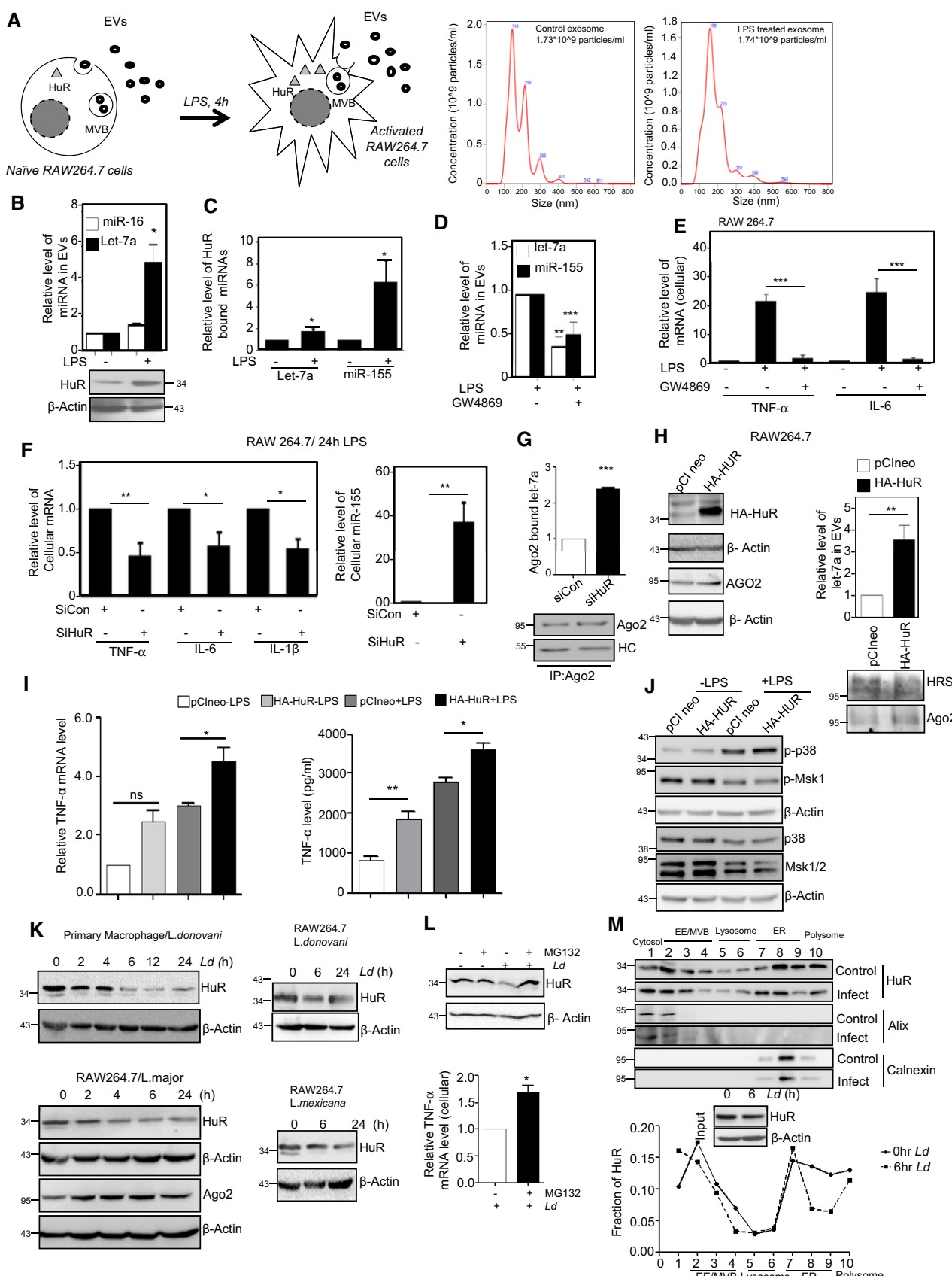

**Figure 5.**

**Figure 5. HuR-mediated extracellular export of miRNA promotes pro-inflammatory responses in macrophage cells.**

A–C   LPS induced miRNA export from mammalian macrophage cells. A schematic diagram of the experiments has been described in left panel of (A). Characterization and quantification of exosomes/EVs derived from control and LPS-activated RAW 264.7 cells by nanoparticle tracking analysis (NTA) done for isolated exosomes/ EVs. Relative size distributions are shown (A, right panel). Levels of HuR and exosomal miRNA isolated from control and 24 h of LPS-treated RAW 264.7 cells. RNA content was normalized against total exosomal proteins (B). β-Actin was used as loading control for HuR Western blot. Binding of miRNAs with HuR after LPS exposure was measured. Amount of RNA was normalized against HuR content. Anti-GFP antibody was used for immunoprecipitation control (C). Values are mean ± s.e.m. and *n* = 3.

D   Effect of GW4869, the inhibitor of exosome/EV-mediated miRNA export, on miRNA content of EVs released by LPS-stimulated cells. The level of miRNAs, let-7a and miR-155 was measured in exosomes (EVs) released from LPS-activated cell after GW4869 treatment. Values for EVs from naïve macrophages without GW4869 and LPS treatment were set as unit. Values are mean ± s.e.m. and *n* = 3.

E   Effect of GW4869 on cellular cytokine levels in LPS-treated cells. The effect of GW4869 on cellular expression of pro-inflammatory cytokines IL-6 and TNF-α in LPS-activated macrophages was measured and normalized against 18S rRNA. Values in naive RAW 264.7 cells without GW4869 and LPS treatment were set as unit. Values are mean ± s.e.m. and *n* = 3.

F, G   Effect of siRNA-mediated downregulation of protein HuR on cellular expression of pro-inflammatory cytokines. In the left panel (F), effect of siRNA treatment on cellular cytokine mRNA levels in LPS-stimulated RAW 264.7 cells is shown. Relative levels measured against 18S rRNA are plotted. The effect of HuR depletion on cellular miR-155 content has been measured, and relative values normalized against U6 snRNA are plotted (right panel, F). The amount of let-7a bound to Ago2 after LPS stimulation in the presence and absence of siHuR has been calculated, and relative values have been normalized to immunoprecipitated Ago2 (G). Values are mean ± s.e.m. and *n* = 3.

H, I   Effect of HuR expression on inflammatory responses in macrophage cells. Effect of ectopic expression of HuR on let-7a content of EVs in naive macrophage cells (H, right panel). HA-HuR expression was checked by Western blot (H, left panel). The effect of HA-HuR expression on production of pro-inflammatory cytokine TNF-α in RAW 264.7 cells was measured. mRNA and protein level of TNF-α was quantified. Values obtained with pCINeo expression and without LPS treatment were taken as unit (I). Transfection was done either with pCIneo (control plasmid) of HA-HuR expression construct, and their effect on cellular pro-inflammatory cytokine mRNA levels in RAW64.7 cells was determined. Values in control set were taken as unit. Values are mean ± s.e.m. and *n* = 3.

J   Effect of HA-HuR expression on p38-mediated activation of downstream signalling events. The levels of p-p38 and p-MSK1 and non-phosphorylated version of the p38 and MSK1/2 have been monitored by Western blotting done with cell extracts prepared from HA-HuR and control plasmid transfected, untreated and LPS-treated RAW 264.7 cells. β-Actin blot was used as loading control.

K–M   Downregulation of HuR in *Ld*-infected macrophages. HuR levels were monitored against time in *Ld*-infected primary macrophage cells (PEC) from BALB/c mice (K, upper left panel) as well as in *Ld*-infected RAW 264.7 cells (K, upper right panel). HuR levels were also monitored against time in *L. major*-infected RAW 264.7 cells (K, lower left panel) and *L. mexicana*-infected RAW 264.7 cells (K, lower right panel). Effect of proteasomal inhibitor MG132 treatment on HuR protein levels in control and 6 h of *Ld*-infected RAW 264.7 cells. In parallel assays, levels of pro-inflammatory cytokine TNF-α were measured in *Ld*-infected cells either with no treatment or pre-treated with MG132 (L). Values are mean ± s.e.m. and *n* = 3. The isotonic cell lysates prepared from 6-h *Ld*-infected cells were analysed on an OptiPrep^R density gradient, and HuR levels in individual fractions were determined. Input samples were monitored for HuR downregulation by Western blot. The presence of Alix (marker for early endosomes) and Calnexin (marker protein for ER) were used to confirm separation of organelles during density gradient fractionation. Loss of HuR from ER-associated fraction has been noted along with their reduction in total lysate after *Ld* infection (M upper panel). Densitometric analysis of individual fractions was done, and the respective values were plotted (M, lower panel).

Data information: HC: heavy chain. In all experimental data, ns: non-significant and *, ** and *** represent *P*-value of < 0.05, < 0.01 and < 0.001, respectively, calculated by Student's *t*-test. For statistical analysis, all experiments are done three times. Exact *P*-values against each experimental set are presented in the Appendix Table S2. Positions of molecular weight markers are marked and shown in the Western blots used in different panels.

Source data are available online for this figure.

from naive and activated macrophage did not differ significantly (Fig 5A, *right panel*). Interestingly, export of miRNA let-7a also got facilitated upon LPS exposure and was found to be HuR-associated (Fig 5B and C). Following the notion that the miRNA export from macrophage cells is responsible for elevated pro-inflammatory response, we noted decreased miRNA export and reduced expression of pro-inflammatory cytokine mRNAs in LPS-stimulated macrophages pre-treated with GW4869, the inhibitor of exosome-mediated miRNA export in mammalian cells [Fig 5D and E (Kosaka *et al*, 2010)]. We have also noted similar decrease in cytokine production upon LPS treatment in macrophages treated with siHuR compared to those treated with siCon. This was accompanied by increased cellular levels of miR-155 that HuR otherwise binds and facilitates its export from LPS-treated cells (Fig 5F). Interestingly let-7a uncoupling from Ago2 that happens in LPS-treated cells also got retarded in cells depleted for HuR (Fig 5G). In this context, miR-155 has been reported to target a TLR pathway adaptor protein Myd88 (Tang *et al*, 2010; Bandyopadhyay *et al*, 2014). Myd88 knock-down alters TNF-α, IL-1β and IL-6 at various conditions suggesting a direct relation between miR-155 and cytokine production mediated by Myd88 (Morandini *et al*, 2013; Lin *et al*, 2015). This rather suggests EV-mediated export as a probable fate of the Ago2-decoupled miRNAs during early hours of LPS stimulation.

Ectopic expression of HuR does not have any major effect on Ago2 expression but increases EV-mediated miRNA export from cells expressing it (Fig 5H). Consistent with the idea of HuR-mediated miRNA export as the major mechanism for balancing inflammatory responses, HA-HuR expression in macrophage has been found to be both necessary and sufficient to cause elevated expression of pro-inflammatory cytokines such as TNF-α both at protein and RNA levels in macrophage cells (Fig 5I) and was accompanied by enhanced miRNA-Ago2 uncoupling noted in LPS-treated cells (unpublished data). However, ectopic expression of HuR did not show any major change in expression of cell signalling components that is known to get upregulated in macrophage upon LPS treatment (Fig 5J). This suggests, instead of a transcriptional surge governed by p38 MAPK-dependent activation of NF-kB controlled pro-inflammatory cytokines, the stabilization of mRNAs due to export of repressive miRNAs by HuR could be necessary and sufficient for the observed pro-inflammatory response associated with HA-HuR expression. Therefore, HuR, here identified as an inducer of pro-inflammatory response, causes upregulation of pro-inflammatory cytokine mRNA levels possibly by facilitating export of cytokine repressor miRNAs. However, as certain cytokines also have AU-rich elements in their 3′-UTR, stabilization of these mRNAs by HuR may also contribute in a dual manner to the high levels of these mRNAs detected in HuR-expressing cells.

### *Leishmania* downregulates HuR to ensure PP2A expression and anti-inflammatory response

The effect of HuR on cytokine mRNAs and miRNAs favours the pro-inflammatory response of the macrophage. It has been observed earlier that *Leishmania* affects the accumulation of miRNPs in the ER-associated fraction that cannot be exported out in *Ld*-infected cells (Chakrabarty & Bhattacharyya, 2017). On the other hand, during pro-inflammatory response, miRNAs are decoupled from Ago2 due to its phosphorylation and get exported out of the cells possibly in a HuR-dependent manner [(Mukherjee *et al*, 2016) Fig 5D and F]. The role of HuR is also evident on miRNA uncoupling from Ago2 (Mukherjee *et al*, 2016; Poria *et al*, 2016) and Fig 5G]. Therefore, the parasite could ensure a retarded miRNA export and poor pro-inflammatory response by targeting HuR.

To support this concept, we determined the expression status of probable HuR target mRNAs in *Ld*-infected human macrophages (Lebedeva *et al*, 2011; Mukherjee *et al*, 2011; Lu *et al*, 2014b). Certain HuR targets (56 mRNA) that could be upregulated on HuR knock-down were also found to be upregulated in *L. donovani*-infected human macrophages, and similarly, certain HuR targets (16 mRNA) that were found to be downregulated on HuR knock-down were also found to show reduced expression in *L. donovani*-infected isolated human macrophages (Appendix Table S1 and Fig S1).

To score the effect of *Ld* infection on cellular HuR levels, we performed a time course experiment with *Ld*-infected RAW 264.7 cells and followed HuR levels after *Ld* infection (Fig 5K). The HuR get cleaved in *Leishmania*-infected macrophage cells. *Leishmania major* and *L. mexicana* also showed HuR level reduction in PEC or RAW 264.7 cells upon infection. The decrease in HuR level could be blocked by inhibiting proteasome function by MG132 (Fig 5L *upper panel*). Blockage of proteasomal degradation of HuR also rescued the pro-inflammatory cytokine TNF-α level in *Ld*-infected RAW 264.7 cells (Fig 5L *bottom panel*). Interestingly, the majority of HuR was lost after 6 h of infection and the loss of HuR happens primarily from the endoplasmic reticulum (ER) and polysome attached fraction of *Ld*-infected cells (Fig 5M). Overall, these results suggest that the primary reason of HuR degradation in *Ld*-infected cells is to reduce pro-inflammatory cytokine production.

The degradation of HuR was also documented when RAW 264.7 cell lysate was incubated with *Ld* lysate (SLA) but that could be blocked by Zn$^{2+}$ chelator *ortho*-phenanthroline (OPT) (Fig EV3A). These data support the involvement of a *Leishmania*l Zn-metalloprotease in cleavage of HuR in infected cells. GP63 is the most prominent Zn-metalloprotease already identified for its role on cleavage of Dicer1 in *Ld*-exposed hepatocytes (Ghosh *et al*, 2013). We tested the purified GP63 and found it to target the cleavage of recombinant His-tagged HuR (data not shown) or cellular HA-HuR in RAW 264.7 cell lysate (Fig EV3B–D). This could be inhibited by Zn$^{2+}$ chelator OPT (Fig EV3E). These data suggest that *Leishmania*l GP63 cleaves cellular HuR protein to ensure robust infection as well as downregulation of pro-inflammatory cytokine production in invaded macrophages. To make a direct relation of HuR cleavage and GP63 protein of *Ld*, we reconstituted liposomes with purified GP63 to treat RAW 264.7 cells to find that GP63 is sufficient to cause the degradation of HuR but not other proteins in GP63 liposome-treated cells (Fig EV3F–I). In multiple experiments, we detected HuR and known GP63 target Dicer1 to get downregulated without much change in β-actin level upon GP63-containing liposome treatment (Fig EV3I).

### PP2A and HuR have common pathways to affect the immune response

Both in LPS-treated and in *Leishmania*-infected macrophage cells, PP2A and HuR expressions were regulated to control miRNA activity. During *Ld* infection, we have noticed increased production of PP2A. However, we observed a decrease in PP2A mRNA levels that could account for a decreased PP2A protein in HA-HuR-expressing cells (Fig 6A–C). In the infection context, the *Ld* infection induced upregulation of PP2A both at protein and mRNA levels (unpublished data) and enhanced the miRNA-mediated repression of pro-inflammatory cytokines. HuR, the negative regulator of PP2A, was noted to be downregulated by the pathogen, possibly to ensure robust anti-inflammatory response observed in infected macrophage cells. Therefore, it is likely that there are reciprocal ways of regulation on inflammatory response that both HuR and PP2A ensure by inversely regulating miRNP machineries targeting the pro-inflammatory cytokines in host cells. We have performed pathway analysis to identify probable regulatory mechanisms between HuR and PP2A (PPP2CA) and determined the common targets that both HuR and PP2A regulate in mammalian cells. Herein, we could identify that

**Figure 6. Inverse regulation of pro-inflammatory pathways by miRNA regulators HuR (ELAVL1) and PP2A (PPP2CA) in mammalian macrophage cells.**

A–C  Interaction routes between HuR (ELAVL1) and its target gene PP2A (PPP2CA). Alternate mechanism or regulatory steps that could be involved in establishing the regulatory relationship between HuR (ELAVL1) and its target genes have been investigated. A brief outline of possible alternate routes of regulation between HuR (ELAVL1) and PP2A (PPP2CA) via intermediates at the protein–protein interaction, cellular signalling or by miRNA has been outlined in panel A. In panels B and C, effect of HA-HuR expression on PP2A both at protein and RNA levels was scored. Values are mean ± s.e.m. and $n = 3$.

D–F  Expression profiles of HuR, PP2A and miRNA let-7a are connected to IL-6 expression in RAW 264.7 cells. Data obtained from experiments described in previous figures are summed up to plot the changes in IL-6 mRNA levels against time of LPS treatment along with changes in PP2A, miRNA let-7a and HuR (D). Change in expression of IL-6 is connected to changes in let-7a concentration. The mathematical equation fitting the curve is shown here. Y represents concentration of IL-6, and X represents miRNA let-7a concentration. A1 is the initial IL-6 levels, and A2 is the changed level. LogX0 is the mid-point of value 0.34, a concentration of let-7a where IL-6 expression is reduced to half (E).The variations of IL-6 expression with the altered level of PP2A (black) and HuR (red). For both cases, the data were fitted to an equation, which is shown above. The parameter $m$ denotes the expression level of PP2A/HuR which corresponds to the maximum IL-6 expression; $\omega$ denotes the width of the distributions for both cases. For PP2A and HuR, the values of $m$ were found to be 0.6 and 0.9, respectively, and A is a constant (F).

Data information: In all experimental data, ns: non-significant and * and ** represent *P*-value of < 0.05 and < 0.01, respectively, calculated by Student's *t*-test. Exact *P*-values against each experimental set are presented in the Appendix Table S2. Positions of molecular weight markers are marked and shown in the Western blots used in different panels.
Source data are available online for this figure.

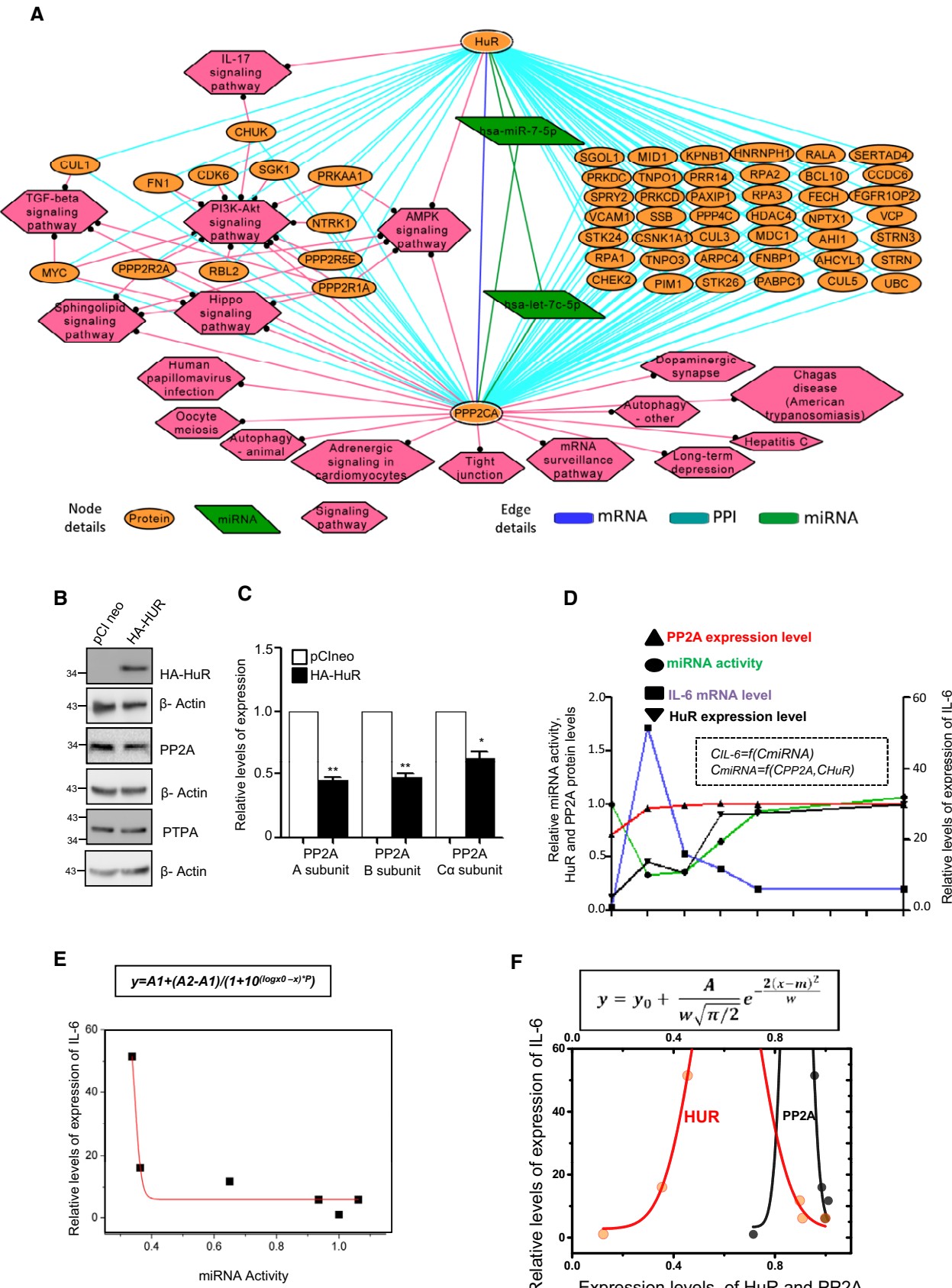

Figure 6.

PP2A (PPP2CA) is directly regulated by HuR at the mRNA level and indirectly regulated via multiple miRNAs (Fig 6A). Further, PP2A (PPP2CA) and HuR (ELAVL1) are commonly involved in regulating inflammatory responses in immune cells.

Using a candidate approach to verify this relation, we plotted the expression level changes of IL-6 mRNA, an important cytokine that is expressed in activated macrophage and regulated by let-7a miRNA (Mazumder *et al*, 2013), and followed the changes in its level in LPS-activated macrophage over time along with changes in miRNA let-7a (known to target IL-6), HuR and PP2A. Both HuR and PP2A control miRNA activity reciprocally and hence should influence the IL-6 expression level inversely (Fig 6D). IL-6 mRNA and miRNA let-7a were found to be in an inverse relationship. Relative changes in IL-6 mRNA levels are linked to let-7a activity as evident in the curve obtained, and the changes in their expression levels are depicted by the equation noted in Fig 6E. By connecting the LPS-mediated changes in IL-6 levels against concentration of PP2A and HuR, we have identified the unique relationship between HuR and PP2A. Mathematical modelling and curve fitting for IL-6 mRNA levels were done in LPS-stimulated macrophages. The equation fitting the data can explain the changes of IL-6 expression as a function of HuR and PP2A concentrations that resulted in a biphasic pattern of IL-6 regulation (Fig 6F). At lower concentration levels, a fluctuation of HuR can influence the IL-6 mRNA level prominently while at a higher concentration PP2A dominates over HuR to control IL-6 mRNA levels. It suggests a mutually exclusive effect of HuR and PP2A on IL-6 cytokine expression which is possibly governed by the action of these proteins on cellular miRNPs (Fig 6D–F).

### HuR can counter PP2A-mediated anti-inflammatory response in *Ld*-infected cells and rescue infection *in vivo*

In the subsequent experiments, we have tested the importance of this biphasic regulation of cytokine expression by HuR and PP2A via miRNPs. As HuR is getting targeted during *Ld* infection, we were interested to test the effect of restoration of HuR expression on *Ld* infection. The number of internalized parasites in HA-HuR-positive cells was notably low compared to the neighbouring macrophage cells not expressing HA-HuR (Fig 7A and B). With HA-HuR expression, we also noted increased expression of TNF-α and IL-1β and reduced expression of IL-10 in HA-HuR-expressing RAW 264.7 cells (Fig 7C). We wanted to see the effect of ectopic expression of HA-HuR on levels of infection *in vivo*. In infected mouse liver also, the level of HuR drops upon infection similar to what was observed in RAW 264.7 cells upon *Ld* infection (Figs 7D and 5K). Similar drop of HuR level in mouse liver could be documented in animals injected with liposomes containing leishmanial GP63 protein (Fig 7E). However, upon the tail vein injection of HA-HuR expression plasmid in infected animals, the expression levels of cytokines IL-10 dropped while the TNF-α expression had increased in liver, and interestingly, infection levels measured by *Ld* kinetoplastid DNA content dropped in mouse liver (Fig 7F–I). No detectable HA-HuR expression was noted in spleen of animals injected with HA-HuR plasmid through tail vein injection. Incidentally, no change in pro-inflammatory cytokine TNF-α could also be detected in spleen (Fig 7J and K). This was of no surprise as tail vein injection route is primarily used to express plasmid encoding genes predominantly in liver tissue (Ghosh *et al*, 2013).

### Robust effect of drug-resistant *Leishmania* *Ld*R on host PP2A expression can be reversed by simultaneous expression of HuR protein and PP2A inhibition

To combat visceral leishmaniasis or *kala-azar*, organic pentavalent antimonials were introduced in the Indian subcontinent almost nine decades ago with dramatic clinical success (Brahmachari, 1922). However, with time, resistance to the drug developed in Bihar, India, that leads to cessation of its further use in the Indian subcontinent. The antimonial drug-resistant form of the *Ld* parasite (*Ld*R) is known for increased IL-10 production upon infection. This is also coupled with stronger reduction of pro-inflammatory cytokine production in the host (Mukherjee *et al*, 2013). Overall, the antimony-resistant *Ld* constitutes an unique example and a model of drug-resistant pathogens with traits of increased fitness and aggressive infection (Mukhopadhyay *et al*, 2015). We used one such strain BHU569 (Mukhopadhyay *et al*, 2011) to test the effect of *Ld*R on the production of HuR and PP2A in RAW 264.7 cells.

Compared to drug-sensitive (*Ld*s) Ag83 strain of *Ld*, the drug-resistant BHU 569 strain (*Ld*R) showed increased IL-10 expression with low TNF-a expression and strong induction of ERK phosphorylation in host cells. It also showed increased drug resistance-associated protein MDR1 expression in RAW 264.7 cells (Fig 8A–D). We also noticed that *Ld*R had an equivalent effect on HuR and a stronger effect on PP2A levels compared to control drug-sensitive *Ld*S Ag83 strain (Fig 8E). Interestingly, unlike what happened to *Ld*S Ag83 strain, HA-HuR expression in RAW 264.7 cells had no effect on internalized parasite number or expression of pro-inflammatory cytokines upon infection with *Ld*R BHU569 strain (Fig EV4). This suggests a strong anti-inflammatory response that *Ld*R raises in RAW 264.7 cells is possibly due to high PP2A expression. The high PP2A effect cannot be counteracted alone by expression of the pro-inflammatory response inducer HuR. However, the mild effect of blockage of PP2A activity by OA on increased IL-1β and TNF-α expression was augmented by the ectopic expression of pro-inflammatory immune response stimulator protein HA-HuR in *Ld*R BHU569-infected cells (Fig 8F). This also holds true in mouse model of infection with *Ld*R. In BALB/c mice infected with *Ld*R, we detected no change in internalized parasite number or cytokine expression in mouse liver by either HuR expression or PP2A inhibition alone. However, like in *ex vivo* experiments simultaneous targeting of pro-inflammatory pathway by HuR expression and PP2A inhibition increased pro-inflammatory response significantly and ensured clearance of *Ld*R from infected mice liver (Fig 8G–I). During *Ld*R infection, the balance should have shifted to anti-inflammatory pathways due to HuR lowering and PP2A upregulation (Fig 8J). However, the reciprocal action of HuR and PP2A in mammalian macrophages in regulation of immune response through their action on miRNP machineries could be utilized to cure infection by *Ld*R when both pathways are targeted.

## Discussion

The expression of cytokines determines the activation status of a macrophage, and the expression of pro-inflammatory cytokines like IL-1β, TNF-α or IL-6 signifies the pro-inflammatory activation state

of the macrophage, whereas increased expression of IL-10 signifies the anti-inflammatory state of the cells. Switching to anti-inflammatory pathway is dominated by suppression of pro-inflammatory cytokine production both at pre- and post-transcriptional stages

(Mosser & Edwards, 2008). The repressive action of miRNAs that targets the pro-inflammatory cytokine encoding mRNAs is considered as the prerequisite for the switching to anti-inflammatory stage (Squadrito et al, 2013).

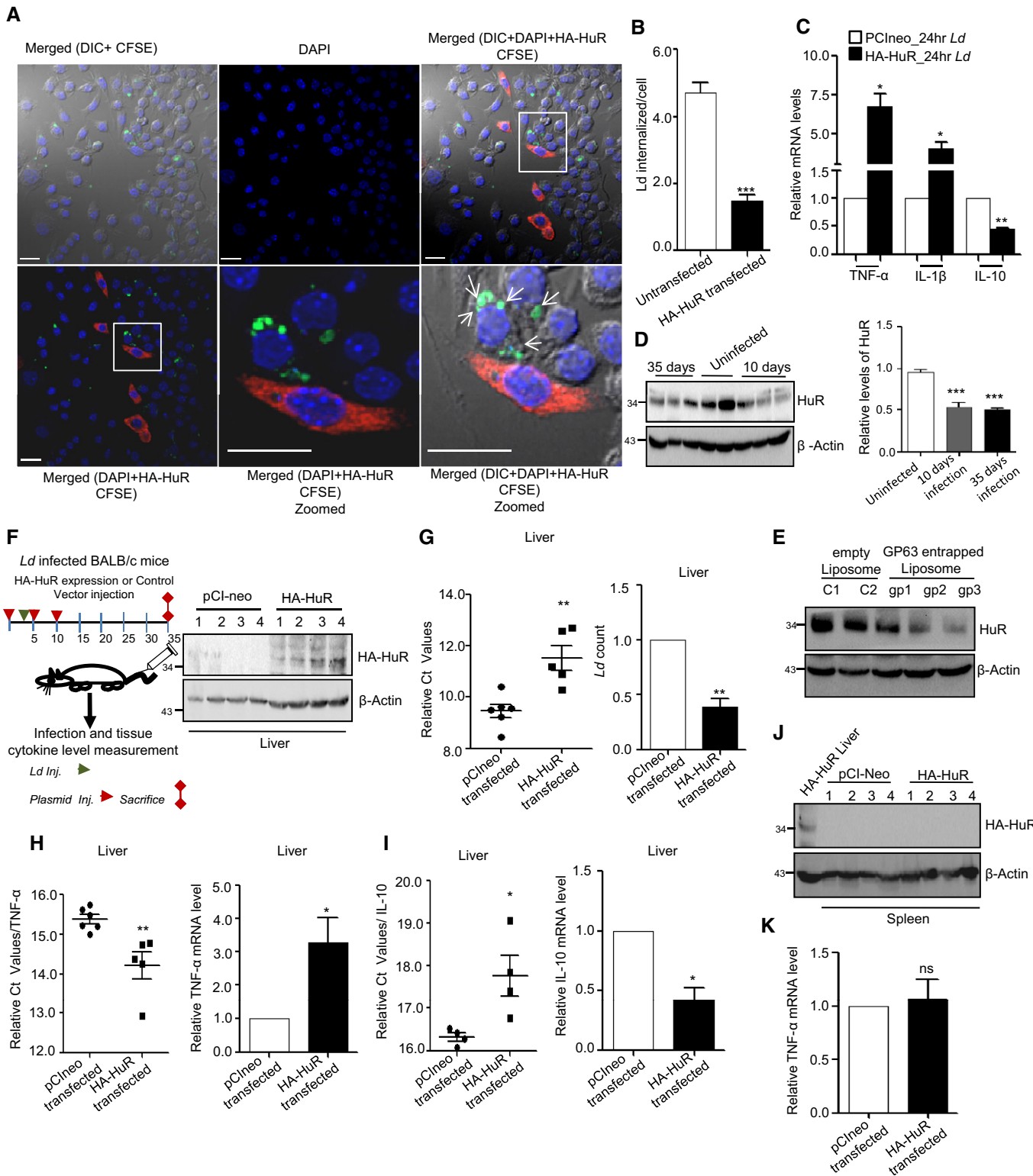

Figure 7.

◄

**Figure 7. *Leishmania donovani* infection could be counteracted by ectopic expression of HuR.**

A–C   Effect of HA-HuR expression on *Ld* infection of RAW 264.7 cells. Effect of HuR expression on internalized parasite number and cytokine expression in RAW 264.7 cells. HA-HuR was transfected to RAW 264.7 cells, and infection was given for 24 h at a host to parasite 1:10 ratio. *Ld* was stained with CFSE dye, and parasite internalization was detected by counting the CFSE-positive structures inside the cells. HA-HuR was immunostained with anti-HA antibody and detected by secondary antibody tagged with Alexa$^R$564 (A). Zoomed part of the merged picture are used to show the internalized parasite. Internalized parasites present in HA-HuR-positive cells were counted and compared against untransfected cells without HA-HuR expression (B). Levels of expression of different cytokines are measured in control or HA-HuR-expressing RAW 264.7 cells infected with *Ld* (C). Scale bar 20 μm. Values are mean ± s.e.m. and n = 3.

D, E   Effect of *Ld* infection on the HuR expression in mouse liver. BALB/c mice of 6 weeks of age were infected with *Ld* via cardiac puncture. Animals were sacrificed after subsequent days; protein was extracted from liver tissue followed by Western blot analysis for HuR. β-Actin was used as loading control (D, *left panel*). Each lane have extracts of livers from individual animal. In each case, HuR level was normalized by respective β-Actin bands (D, *right panel* mean ± s.e.m., n = 5). Animals were injected with GP63-containing liposome (gp1–3) or empty liposome (C1-2), and after 24 h of injection, HuR level was checked against β-Actin used as loading control (E).

F–I   Effect of expression of HA-HuR on *Ld* infection in mice liver. Scheme for animal experiments done by expressing HA-HuR in mouse liver is shown. HA-HuR level was detected in the liver tissue by Western blot analysis using anti-HA antibody (F). Parasite load in liver tissues was estimated by measuring *Ld* DNA in infected tissue using specific primer for *Ld* minicircle kDNA. Values are mean ± s.e.m. and n = 5 (G). TNF-α (values are mean ± s.e.m. and n = 5) and IL-10 (values are mean ± s.e.m. and n = 4) mRNA levels were checked from RNA isolated from liver tissues, and 18S rRNA normalized $C_t$ values were plotted (H and I, left panels). Relative fold change was calculated by $\Delta\Delta C_t$ method and the values plotted for *Ld* count, TNF-α and IL-10 mRNA levels [G mean ± s.e.m. (n = 5), H mean ± s.e.m. (n = 6) and I mean ± s.e.m. (n = 4), right panel].

J, K   Effect of HA-HuR plasmid tail vein injection on mice spleen. Levels of HA-HuR were undetectable in mice spleen after HA-HuR expression plasmid was injected. Sample from one liver tissue of HA-HuR injected group was used as a positive control of HA-HuR expression in mouse liver after tail vein injection of HuR encoding plasmid (J). Simultaneously, TNF-α mRNA levels were checked in spleen total RNA (K). Values are mean ± s.e.m. and n = 4.

Data information: In all experimental data, ns: non-significant and *, ** and *** represent *P*-value of < 0.05, < 0.01 and < 0.001, respectively, calculated by using Student's *t*-test. For statistical analysis, all experiments are done three times. Exact *P*-values against each experimental set are presented in the Appendix Table S2. Positions of molecular weight markers are marked and shown in the Western blots used in different panels.
Source data are available online for this figure.

miRNA-mediated regulation controls varieties of genes including several cytokine encoding genes (Palanisamy *et al*, 2012). It has also been observed that reversible uncoupling of miRNA from Ago2 ensures switching of miRNA activity in mammalian cells (Mazumder *et al*, 2013; Patranabis & Bhattacharyya, 2016). It has also been noted that *Ld* infection controls expression of several miRNAs including those that act as repressor of pro-inflammatory cytokine mRNAs (Chakrabarty & Bhattacharyya, 2017). In infected cells, a general increase in miRNA stability was observed and increased miRNAs were found to be associated with the ER attached fraction. The increased stability of miRNAs can be attributed to decreased miRNA export. HuR is known to be involved in exosome-mediated export of miRNAs. In cells, infected with *Ld*, we noted a decrease in HuR levels. However, the role of PP2A is important also in *Ld*-infected macrophage where its high expression ensures restriction of Ago2 in its dephosphorylated and miRNA-bound form. HuR expression is both necessary for unbinding of miRNAs and their extracellular export. The pathogens which reside within the macrophage tame the pro-inflammatory response and achieved this by targeting pro-inflammatory cytokine mRNA production both at pre- and post-transcriptional level. It has been reported earlier that induction of PP2A phosphatase is required for inactivation of p38/MAPK pathways, and thus, PP2A prevents not only the expression of pro-inflammatory cytokine mRNAs at transcriptional levels by targeting NF-kB pathway and related signalling events (Kar *et al*, 2010) but also it helps to keep Ago2 in its dephosphorylated miRNA-bound active form to repress the existing mRNAs. In this process, inactivation of p38 pathways possibly ensures the down-regulation of nucleocytoplasmic shuttling of HuR shown to be essential for HuR action on its targets (Tiedje *et al*, 2012) and to restrict subsequent miRNA turnover and export by cytoplasmic HuR. Opposite event happens during LPS activation phase where the cytoplasmic HuR promotes the export of miRNAs and ensures the miRNA recycling prerequisite for restoration of miRNA-mediated

repression of excess cytokine mRNAs otherwise detrimental for activated macrophage cells. HuR has a negative role on PP2A mRNA expression and as expected should be useful to downregulate PP2A expression to ensure maximum activation of macrophage during LPS exposure. However, the HuR-mediated downregulation mechanism of PP2A is not clear. On the contrary, PP2A level increases during late phase of LPS activation, and thus, it can be concluded that PP2A may dominate over HuR during late phase of activation. This is also supported by the mathematical model of macrophage activation. However, HuR gets downregulated during *Ld* invasion of macrophage. This possibly ensures a robust downregulation of pro-inflammatory response associated with *Ld* infection.

From the mechanistic angle, the contribution of subcellular structures and compartments involved in immune response of *Ld* invaded macrophages is an underexplored question, and it is not clear why the majority of miRNPs are usually associated with ER while the PP2A and phosphorylated Ago2 get accumulated in endosomal membrane (Patranabis & Bhattacharyya, 2016; Chakrabarty & Bhattacharyya, 2017). Perhaps, by ensuring compartmentalization of miRNA loading and miRNP phosphorylation to ER and endosomes, respectively, the spatio-temporal control of macrophage activation process is achieved to ensure a robust regulation at post-translational level.

In the context of infection by drug-resistant pathogen, these findings have some direct implications. Interestingly, 78% of the recent clinical isolate of *L. donovani (Ld)* from endemic zone of Bihar show resistance to organic pentavalent antimonials (Mukhopadhyay *et al*, 2011). The resistant parasite ($Ld^R$) induces production of disease-promoting cytokines like IL-10 from macrophages (Mukherjee *et al*, 2013). It also upregulates multidrug resistance-associated protein-1 (MDR-1) in infected macrophages (Mukherjee *et al*, 2013). These attributes are essentially absent when macrophages are infected with antimony-sensitive *Ld* parasites ($Ld^S$). The mechanism by which $Ld^R$ induce aggressive infection is not clearly known. $Ld^R$ upregulates miR-466i in macrophages that binds to 3′-UTR causing

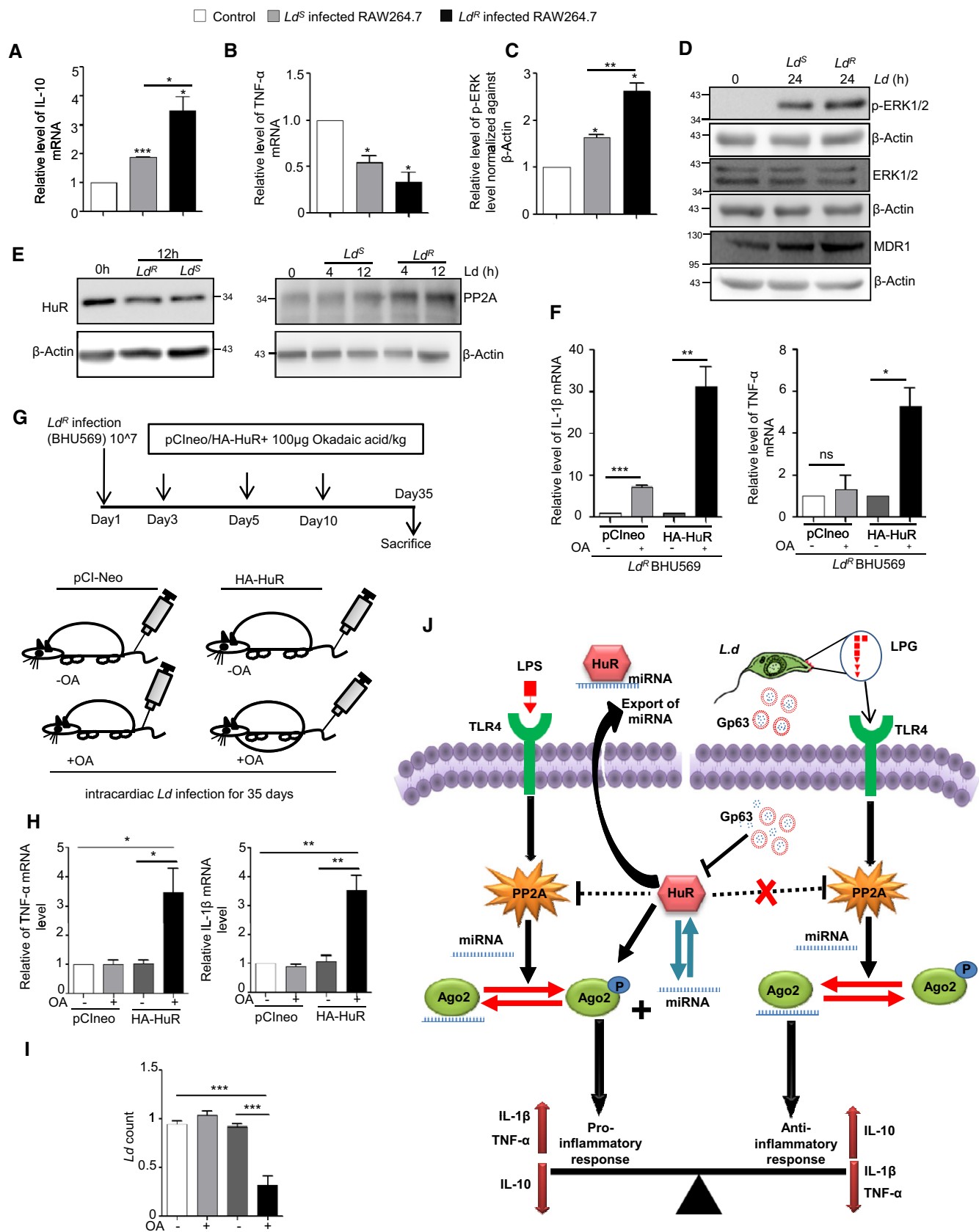

**Figure 8.**

◀

**Figure 8. Requirement of simultaneous activation of HuR and deactivation of PP2A driven processes in mammalian macrophage to prevent infection by antimonial drug-resistant *Leishmania* parasite.**

A–D  Effect of antimonial drug-resistant and drug-sensitive forms of the *Ld* (*Ld^R* and *Ld^S*) on cellular cytokine levels as well as on phospho-ERK and MDR1 levels. RAW 264.7 cells were infected with antimony-sensitive (*Ld^S*-Ag83) and antimony-resistant (*Ld^R*-BHU569) form of *Ld* for 24 h along with uninfected cells kept as control. IL-10 (A) and TNF-α (B) were measured at mRNA level by qRT–PCR after infection (mean ± s.e.m. and *n* = 4). p-ERK, ERK and MDR1 levels were also measured by Western blot (D). Densitometry analysis was done for p-ERK level by ImageJ, and relative values were plotted (C). β-Actin was used as loading control. Values are mean ± s.e.m. and *n* = 3.

E  Effect of antimony-sensitive and antimony-resistant form of the *Ld* on HuR and PP2A. RAW 264.7 cells were infected with *Ld^S* and *Ld^R* for 12 h, and HuR level was measured by Western blot (left panel). PP2A level was also measured by Western blot after 4 h and 12 h of *Ld^S* and *Ld^R* infection (right panel). β-Actin was used as loading control.

F  Effect of ectopic expression of HuR and PP2A inhibitor OA treatment on *Ld^R* infection of macrophage cell. RAW 264.7 cells were transfected with HA-HuR or pCIneo followed by 2 h of pre-treatment with OA. These cells were then infected with *Ld^R* for 24 h. Pro-inflammatory cytokines TNF-α (right panel; mean ± s.e.m., *n* = 3) and IL-1β (left panel; mean ± s.e.m. and *n* = 4) were measured at mRNA level, and relative values are plotted. Values without OA treatment were set as control.

G–I  Effect of HuR over expression on *Ld^R* infection in mice liver. A scheme for animal experiment done with *Ld^R* infection followed by HA-HuR expression and OA treatment is shown (G). Pro-inflammatory cytokines TNF-α (left panel) and IL-1β (right panel) levels were measured at mRNA level in liver (H). Parasite load was measured in infected liver tissue by using specific primer against kinetoplastid minicircle DNA of *Ld* (I). Values are mean ± s.e.m. and *n* = 6.

J  The model depicts inter-regulatory balance between PP2A and HuR in controlling pro-inflammatory and anti-inflammatory response by altering Ago2 phosphorylation–dephosphorylation cycle and exosomal export of miRNAs in mammalian macrophage cells. Left half of the model shows lipopolysaccharide (LPS)-induced PP2A upregulation via TLR4 pathway during late hours of treatment which results in Ago2 dephosphorylation. Right half of the scheme represents *Leishmania*-induced PP2A increase mediated by membrane glycolipid lipophosphoglycan (LPG). Interestingly, the RNA binding protein HuR plays a central role in both the contexts. Inhibitory effect of HuR on PP2A ensures phosphorylated form of Ago2 to dominate which in turn facilitates miRNA unbinding to Ago2 and miRNA export via exosome thereby promoting pro-inflammatory cytokine response. On the contrary, *Ld* induced downregulation of HuR via a zinc-metalloprotease, GP63 which results in PP2A upregulation that facilitates anti-inflammatory response necessary for the parasite survival and proliferation.

Data information: In all experimental data, ns: non-significant and *, ** and *** represent *P*-value of < 0.05, < 0.01 and < 0.001, respectively, quantified by using Student's *t*-test. For statistical analysis, all experiments are done three times. Exact *P*-values against each experimental set are presented in the Appendix Table S2. Positions of molecular weight markers are marked and shown in the Western blots used in different panels.

Source data are available online for this figure.

downregulation of MyD88 and resulting in elevated IL-10/IL-12 ratio (Mukherjee *et al*, 2015). This elevated IL-10 over IL-12 may cause aggressive pathology in BALB/c mice when infected with *Ld^R* as compared to sensitive counterpart. It is also known that *Ld^R* attenuates the therapeutic efficacy of antimonials by inhibiting dendritic cells, the key regulators of anti-leishmanial immune responses (Haldar *et al*, 2010). Drug resistance represents one of the main problems for the use of chemotherapy to treat leishmaniasis. Additionally, it could provide some advantages to *Leishmania* parasites, such as a higher capacity to survive in stress conditions. Further to this, intracellular amastigotes from resistant lines present a higher capacity to survive inside the macrophages than those of the control line. Our results suggest that resistant parasites acquire an overall fitness increase than the sensitive counterpart (Roy *et al*, 2014) and that can be reversed by changing the balance of pro-over anti-inflammatory response controlled by miRNPs and its modulators HuR and PP2A.

The drug-resistant BHU569 strain used in this work is one of the clinical isolates previously characterized in terms of their sensitivity towards anti-leishmanial drug sodium stibogluconate (SSG) in macrophage culture system by Mukhopadhyay *et al* (2011). Also, they had studied a series of genes related to SSG transport in the parasites itself. The resistant parasites were shown to overexpress MRPA gene higher than the sensitive one along with the higher expression of an unique terminal glycoconjugate *N*-acetyl-D-galacto-saminyl on cell membrane of resistant parasites. Higher IL-10 production and higher level of MDR1 expression in macrophages infected with of *Ld^R* strain comparative to the *Ld^s* strains were also reported by them (Mukhopadhyay *et al*, 2011).

Later, the same group revealed the molecular mechanism behind higher IL-10 production and MDR1 expression in case of resistant strain infection which showed that the terminal glycoconjugate

interacts with TLR2/6 which resulted in more IL-10 production via upregulation of phospho-ERK level than the sensitive one (Ag83). The higher MDR1 expression was found to be IL-10-driven (Mukherjee *et al*, 2013).

Here, our observation reveals that the BHU569 strain can produce more IL-10 upon infection via ERK1/2 pathway by higher upregulation of phospho-ERK1/2 level than the sensitive Ag83 strain. In this work, we have found *Ld^R* infection to trigger PP2A level more than that in the *Ld^s* strain. It also produces more IL-10. However, no direct relation has been found between higher IL-10 production and higher PP2A level. One such work reported previously in case of experimentally generated parmomycin-resistant *L. donovani* increased level of MRPA, MDR1 and PP2A was found in the resistant parasites than their sensitive counterparts along with increased IL-10 production upon infection (Bhandari *et al*, 2014). Other groups have reported that a protein Tyrosine phosphatase PTPN3 gene expression increases in cisplatin and doxorubicin-resistant ovarian cancer cells (Li *et al*, 2016). All these previous reports suggest that somehow protein phosphatases are involved in drug resistance phenomenon.

Our observation reveals higher expression of IL-10 and MDR1 along with PP2A upon *Ld^R* infection. Hence correlating with previous reports, it can be assumed that PP2A might be involved in regulation of MDR1 expression and responsible for their increased pathogenicity and drug resistance phenomenon of these antimony resistance parasites. However, our unpublished data indicate similar increase in PP2A in macrophage upon infection with another *Ld^R* strain, and blockage of ERK pathway by chemical inhibitor could retard the PP2A expression in infected cells. This suggests importance of ERK signalling in PP2A expression.

HuR is primarily known to associate with specific upstream/downstream sequences of target mRNAs, for instance, U/AU-rich

sequences either at 3′-untranslated regions (UTRs) or within pre-mRNA introns (Srikantan & Gorospe, 2011). Since RNA-HuR inter-actions have been extensively characterized in PAR-CLIP analysis considering HeLa or HEK293 cells in previous works (Lebedeva *et al*, 2011; Mukherjee *et al*, 2011), we have utilized the data provided to generate a list of HuR target mRNAs that may be regu-lated also in macrophages. During *Ld* infection, HuR expression goes down. Hence, the underlying assumption was that as the binding sites of the target genes would be present in the macrophage cell  mRNA as well, HuR-mediated regulation of target genes' expression would also be altered during *Ld* infection. Subsequently, by determining whether any of the probable HuR targets are dif-ferentially expressed when human macrophages are infected with *Ld*, we could predict 165 likely mRNA targets of HuR that may have a role in macrophages in *Ld* infection scenario. Additionally, consid-ering RNAseq profile of mRNA from HuR (*Elavl1*) knockout murine bone marrow-derived macrophages (Lu *et al*, 2014a) described in the GEO dataset (GSE63199), we could observe that 129 among the previously predicted 165 targets show differences in their expression profile under these conditions as well.

Mechanism by which pro-inflammatory cytokine mRNA expres-sion was upregulated by HuR is not entirely clear. While due to defined AU-rich HuR binding site of TNF-α mRNA can account for its stabilization and expression against miRNA-mediated repression, the situations with other pro-inflammatory factor are elusive. Inter-esting work by other groups has shown previously that HuR regu-lates TNF-α-induced IL-6 production by stabilizing IL-6 mRNA that otherwise gets negatively regulated by another RNA binding protein TTP (Shi *et al*, 2012). Consistent with this finding, other groups have reported LPS-induced p38 pathway mediates phosphorylation of TTP which alters its inhibitory effect on IL-1β mRNA levels (Chen *et al*, 2006; Neuder *et al*, 2009). IL-1β also has binding sites for HuR; thus, HuR directly also can influence its expression. IL-10 mRNA expression was undetectable in HA-HuR-expressing cells, and thus, we could not comment on that.

The immunogenic properties of the EV are dependent on the cargos they carry. The immunostimulatory role has already been reported for liver-specific miR-122 that can be transferred via EVs to neighbouring macrophage cells to get them stimulated (Saha *et al*, 2018; Xu *et al*, 2018). Similar to what has been described in our previous paper where role of HuR in enhancing EV-mediated export of miR-122 has been described, we can attribute a role of HuR in contributing the inflammatory response propagation in that context (Basu & Bhattacharyya, 2014; Mukherjee *et al*, 2016). Therefore, the effect could be very much contextual. In macrophage, HuR by exporting out repressive let-7a or similar miRNAs during early phase of LPS stimulation allows pro-inflammatory cytokines to get expressed and thus ensures pro-inflammatory response to set in HuR-expressing cells. Interestingly, miR-155, a pro-inflammatory miRNA (Squadrito *et al*, 2013), also get exported out from macro-phage expressing HuR. This pro-inflammatory signal is communi-cated to neighbouring naive macrophages to get them stimulated via a secondary pathway. Therefore, the data are not inconsistent with previously published data rather in line with the findings by us and others on both pro- and anti-inflammatory role of EVs. It could be determined by the cargos they carry.

Interestingly, the findings described in this manuscript suggest that the balance of anti- and pro-inflammatory responses is delicately achieved via miRNA-mediated regulation and by targeting the miRNA modulators, in this case HuR and PP2A. One can arbi-trate the immunological response of host macrophage to control parasite infection. This should hold true for other context like in tumour microenvironment or in auto-immune complications where pro- and anti-inflammatory behaviours of the immune cells are context-dependent and may also be regulated in similar manners by separate sets of miRNA modulators and HuR-like factors.

# Materials and Methods

## Chemicals and plasmid constructs

The macrophage cells were stimulated with 1 ng/ml *Escherichia coli* O111:B4 LPS (Calbiochem, La Jolla, CA). For PP2A-specific inhibi-tion, 100 nM okadaic acid (Calbiochem, CA) was used. Exosome secretion blocker GW4869 (Calbiochem, CA) was used at 10 μM concentration. Plasmid information about pRL-con and pRL-3XBulge-Let7a was published previously (Pillai *et al*, 2005) and was a kind gift from Witold Filipowicz. HA-HuR-expressing plasmid was obtained from Witold Filipowicz which was cloned into pCIneo backbone and described previously (Kundu *et al*, 2012). FH-Ago2 and FH-Ago2Y529F were obtained as kind gifts from Gunter Meister. Tet-inducible precursor miR-122 was cloned in pTRE-tight-BI vector (Clontech) and described in previous report (Ghosh *et al*, 2015). All siRNAs were purchased from Dharmacon Inc. (On Target Plus siRNAs).

## Cell culture and reagents

RAW 264.7 cells were grown in RPMI 1640 medium (Gibco) supple-mented with 2 mM L-glutamine, 0.5% β-mercaptoethanol, 1% Pen-strep (Gibco) and 10% heat-inactivated foetal calf serum (complete media). The cell line was obtained from ATCC.

Peritoneal exudate cells (PEC) of BALB/c mice were elicited by 4% starch injected in the mouse peritoneal cavity, and 24 h after starch boost, mice were sacrificed, skin was carefully removed, and peritoneal lavage containing the PEC was extracted with cold 1× PBS. Cells were spin down and resuspended in complete RPMI media. All experiments were done after 48 h of cell isolation. PEC was isolated from adult BALB/c mice from any sex.

For macrophage activation, 1 ng/ml LPS was used. For all okadaic acid (OA) treatment experiments, 100 nm OA was used to pre-treat the cells for 2 h before further treatment either with LPS or infection.

## Parasite infection of macrophage cells

*Leishmania donovani* strain AG83 (MAOM/IN/1083/AG83) was originally obtained from an Indian viceral leishmaniasis ("kala-azar") patient and maintained in golden hamster. To infect RAW 264.7 cells, 2nd–4th passage cultures of *Ld* promastigotes were used in 10:1 ratio for all experiments to infect RAW 264.7 or PEC. Sodium stibogluconate-resistant clinical isolate of *Ld* strain MHOM/IN/09/BHU569/0 was also maintained by similar method.

Stationary phase *L. major* (Friedln strain) and *L. mexicana* (M379) parasites from 3rd to 5th passage were maintained in M199

media supplemented with 10% FBS (Gibco) and 1% Pen-strep solution (Gibco). Infection was given to the RAW 264.7 cells at 1:10 host:parasite ratio. *Leishmania major* was maintained by infecting hamster footpad with $10^7$ parasites; thereby, amastigotes were isolated from footpad after 6–8 weeks of infection.

### Cell transfection

For luciferase assay, 25 ng of RL reporters and 400 ng of FF reporters were transfected using Lipofectamine 2000 following manufacturer's protocol in 12-well format. Transfection of HA-HuR, pCIneo and FH-Ago2 plasmids was transfected by Fugene HD (Promega) in 1:5 ratio (μg of DNA: μl of Fugene HD). For co-transfection of siRNA and plasmids, Lipofectamine 2000 was used.

### Immunoprecipitation

For immunoprecipitation, protein G agarose beads (Life technologies) for endogenous Ago2 or FLAG-M2 agarose beads (Sigma) for exogenous FLAG-tagged Ago2 were washed with 1× IP buffer (20 mM Tris–HCl, pH7.5; 150 mM KCl, 1 mM MgCl$_2$). For endogenous Ago2, beads were blocked with 5% BSA in lysis buffer (1× IP buffer with 0.5% Triton X-100) for 1 h before antibody addition (1:50 dilution) for overnight at 4°C. Cells were lysed in 1× lysis buffer for 30 min at 4°C followed by 10-s sonication for three times with 5-min incubation on ice in between. Lysate was pre-cleared by centrifugation at 16,000 × g at 4°C after lysis. Lysate was then added to the antibody-bound beads for 4 h after which beads were washed with IP buffer for three times at 4°C. Proteins and RNAs were extracted from the beads for further analysis either with 1× SDS sample buffer or Trizol LS (Invitrogen) reagent.

### Exosomes isolation and characterization

Cell culture supernatants were used for exosome isolation. For exosome isolation, cells were grown in exosome-depleted media to prevent any background levels of exosomes present in FBS used to prepare the culture medium. Briefly, the cell supernatant was centrifuged at 3,000 × g for 15 min to remove cell debris. Next, the cell supernatant was collected and centrifuged at 10,000 × g for 30 min. The supernatant that was obtained was passed through 0.22 μm filter units to further remove debris. This was followed by ultracentrifugation of the supernatant at 100,000 × g for 90 min. After ultracentrifugation, the pellet was resuspended in either PBS or complete media for further use.

For nanoparticle tracking analysis (NTA), exosome from $10^6$ cells was resuspended in 1 ml PBS. Exosome was 10-fold diluted, 1 ml of diluted exosome was injected into the sample chamber of nanoparticle tracker (Nanosight NS300), tracking were done, and video was captured.

### Luciferase assay

Control Renilla (RL-con) and/or let-7a reporter RL-3XBulge-let-7a containing three miRNA binding sites was used to check the miRNA let-7a activity. Firefly (FF) reporter plasmid was used for normalization of transfection. The Renilla and Firefly levels were measured using Dual Luciferase Assay Kit (Promega) using manufacturer's

protocol. The ratio of FF normalized RL-con to FF controlled RL-3XBulge-let-7a was used to measure the repression level (Mazumder *et al*, 2013).

### OptiPrep™ density gradient centrifugation

OptiPrep™ (Sigma-Aldrich, USA) was used to prepare a 3–30% gradient in a buffer containing 78 mM KCl, 4 mM MgCl$_2$, 8.4 mM CaCl$_2$, 10 mM EGTA and 50 mM HEPES (pH 7.0) in a 4 ml ultracentrifuge tube. The protocol has been standardized for separation of subcellular organelles as described previously (Ghosh *et al*, 2015). RAW 264.7 cells were washed with PBS and homogenized with a Dounce homogenizer in a buffer containing 0.25 M sucrose, 78 mM KCl, 4 mM MgCl$_2$, 8.4 mM CaCl$_2$, 10 mM EGTA, 50 mM HEPES pH 7.0 supplemented with 100 μg/ml of cycloheximide, 5 mM vanadyl ribonucleoside complex (VRC) (Sigma-Aldrich), 0.5 mM DTT and 1× protease inhibitor (Roche). The lysate was clarified by centrifugation at 1,000 × g for 5 min two times and layered on top of the prepared gradient. The tubes were centrifuged at 110,000 × g for 5 h for separation of gradient on a SW60 Ti rotor. About 400 μl of each fraction was collected by aspiration from the top for subsequent analysis.

### Estimation of total cellular and Ago2-bound mRNA and miRNAs

For miRNA estimation, cellular miRNA levels were detected using TaqMan-based miRNA specific assay kit started with 50 ng of total RNA concentration. U6 was used for normalization. Ago2-bound miRNA was estimated after immunoprecipitation of Ago2 using equal volume of RNA extracted after immunoprecipitation. Value normalized against total amount of Ago2 immunoprecipitated in each case. Real-time PCR was performed with 25 ng of cellular RNA and 200 ng of EV RNA unless specified otherwise, using specific primers for human let-7a (assay ID 000377), human miR-122 (assay ID 000445), human miR-16 (assay ID 000391), human miR-146a (assay ID 000468) and human miR-155 (assay ID 002571). U6 snRNA (assay ID 001973) was used as an endogenous control. The reverse transcription mix was used for PCR amplification with TaqMan® Universal PCR Master Mix No AmpErase (Applied Biosystems) and analysed in triplicates for each biological replicates. The comparative $C_t$ method which typically included normalization, by the U6 snRNA, was used for quantification.

For mRNA estimation, mRNA levels were estimated by SYBR Green-based real-time cDNA estimation using specific primer for respective genes. For that, 200 ng of total RNA was used. For Ago2-bound mRNA estimation, equal volume of RNA extracted after immunoprecipitation was used. Value normalized against total amount of Ago2 immunoprecipitated in each case. Ago2 protein level quantified using ImageJ software done for blots obtained after Western blot analysis of immunoprecipitated Ago2. The comparative $C_t$ method which typically included normalization by the GAPDH mRNA or 18S rRNA was used for quantification.

### Immunofluorescence

Cells were fixed using 4% para-formaldehyde (PFA) for 20 min. Blocking and permeabilization was done by 1% BSA, 10% goat serum and 0.1% triton X-100 for 30 min. *Ld* promastigotes were stained by 1 μM carboxyfluorescein succinimidyl ester (CFSE) dye (Green) for

30 min at 22°C rocking followed by washing with 1× PBS thrice before infecting RAW 264.7 cells. The anti-HA (Rat) and anti-GP63 (Mouse) were used at 1:1,000 dilutions. Secondary Alexa Fluor$^R$ 405 and 568-labelled anti-rat and anti-mouse antibodies were used at 1:500 dilutions. Cells were observed under Zeiss LSM800 confocal microscope. In HA-HuR overexpressed cells, primary anti-HA and Alexa Fluor$^R$ 568 anti-rat secondary antibody was used. For calculating per cent of infected cells, minimum of 100 cells were counted.

### In vitro phosphatase assay

FH-Ago2 was immunoprecipitated from stable transfected HEK293 cells expressing FH-Ago2. The phosphatases were pulled down using specific antibody from RAW 264.7 cell extracts prepared under different reaction conditions. The beads were washed with 1× phosphatase buffer (50 mM Tris–HCl, pH 7.5, 1 mM MgCl$_2$, 100 μM EDTA, 4.5 mg BSA, 1× protease inhibitor) prior to the reaction. FA-Ago2 was eluted and mixed with immunoprecipitated phosphatase containing beads, and volume was adjusted to 100 μl using 1× phosphatase buffer and incubated at 37°C for 30 min before the reaction is stopped with 1× SDS sample dye containing phosphatase inhibitor OA. The reaction was analysed on SDS–PAGE, and quantity of HA-Ago2 and phospho-Ago2 was determined by HA and phosphotyrosine-specific 4G10 antibodies, respectively.

### SLA and GP63 purification

Soluble Leishmania antigens (SLA) and GP63 purification were done by following the published protocol (Bhowmick et al, 2008; Ghosh et al, 2013). Briefly, $1 \times 10^{10}$ promastigotes cells were used as a starting material, and after series of thorough lysis, ultracentrifugation and affinity-based purification, GP63 was successfully purified.

### In vitro HuR cleavage assay

RAW 264.7 cell lysate was incubated with Leishmania-derived soluble Leishmania antigen (SLA) or purified GP63 at 37°C for 30 min. The reaction was terminated using 5× SDS buffer. Ortho-phenanthroline was used as GP63 inhibitor. SLA was pre-treated with 10 μM ortho-phenanthroline for 30 min on ice before the reaction. The reaction was performed in a buffer containing 10 mM Tris–HCl pH 7.5, 100 mM KCl, 1 mM DTT and 1× protease inhibitor (Roche).

### Animal experiments

BALB/c female mice of age 4–6 weeks were infected with $1 \times 10^7$ promastigotes via cardiac puncture maintained and sacrificed after 10$^{th}$ day and 35$^{th}$ day of infection. For the exogenous expression of HA-HuR experiment, mice were injected with 25 μg pCIneo or HA-HuR expression plasmid. On the third day, $1 \times 10^7$ promastigotes were used to infect the mice by cardiac puncture. Booster doses of plasmid were given on the 5$^{th}$ and the 10$^{th}$ day after the 1$^{st}$ day of plasmid administration. All mice were sacrificed on 35$^{th}$ day of infection, and tissues were snap frozen for future use. Approximately 10 mg tissue slice was collected and homogenized using 1× RIPA buffer for protein estimation and Western blotting. TriZol was directly added to the tissue slices, homogenized rigorously for thorough lysis before RNA isolation. DNA isolation was done following

manufacturer's protocol using Promega Wizard DNA isolation Kit. All animals were maintained in individually ventilated cages having sterile air supply and regulated temperature. For isolation, peritoneal macrophage adult female mice were used. All procedures were performed in accordance with a protocol approved Institutional Animal Ethics Committee. All the experimentations were performed according to the National Regulatory Guidelines issued by the Committee for the Purpose of Supervision of Experiments on Animals, Ministry of Environment and Forest, Govt. of India. All the experiments involving animals were carried out with prior approval of the Institutional Animal Ethics Committee. All mice were housed in individually ventilated caging system with 12 h/12 day/night cycle.

### LPG purification and treatment

LPG was extracted was described previously with modifications (Piedrafita et al, 1999). Briefly, parasites were cultured and grown to a density of $10^7$ cells/ml. The parasites were centrifuged at $200 \times g$ for 10 min. Cells were resuspended in 2 ml of chloroform:methanol:water (1:2:0.5, v/v/v) for 2 h at room temperature. The insoluble pellet was used for LPG extraction. De-lipidation of LPG was done with 9% butanol in water for 1 h each time for twice ($2 \times 250$ μl), and the fractions were pooled and vacuum dried. A total of $10^8$ cells were used for LPG extraction. The overall extraction was resuspended in 100 μl of 1× filtered PBS and sonicated before treatment. As LPG was extracted from $10^8$ cells and resuspended in 100 μl of PBS, 1 μl of the extract corresponds to approx $10^6$ cell equivalent of LPG.

### GP63 entrapment to liposome and treatment

GP63 entrapment in liposome was done using a previously described procedure (Ghosh et al, 2013) with slight modifications. Briefly, 12 mg phosphatidylcholine (Avanti Polar Lipids) (100 mg/ml), 8.7 mg cholesterol (Sigma) (100 mg/ml) (1:1.5 of cholesterol to PC molar ratio) and 6 mg n-octyl-β-D-glucopyranoside (Sigma) were dissolved in chloroform. The lipid mixture was coated as a uniform layer in a round glass bottle and kept in the desiccator. The lipid layer was solubilized in PBS containing either 250 μg purified GP63 or in the absence of GP63 and resuspended. The solution was sonicated in a bath sonicator for 15 min on ice water. The liposomes were ultracentrifuged at $100,000 \times g$ at 4°C for 1 h to remove unincorporated protein and lipid particles. The liposomal pellet was resuspended in normal saline for animal or cell treatment. Approximately 20 μg of GP63 entrapped liposome was administered to the animal (BALB/c female adult mice) via intracardiac route. Mice were sacrificed after 24 h of administration. Liver tissues were collected for Western blot analysis. It has been shown by Ghosh et al (2013) that most of the administered liposome gets accumulated in the liver. For RAW 264.7 cell treatment, 1 μg of GP63 entrapped liposome were used for $10^6$ cells.

### Real-time PCR for detection of Leishmania

The real-time detection to estimate Ld count was done as previously described (Nicolas et al, 2002). Briefly, DNA was collected from liver tissue of mice and 200 ng of isolated DNA was used for the

real-time detection. 18S rRNA was used as endogenous control. The *Ld* count was calculated from relative values obtained by $\Delta\Delta C_t$ analysis. Appendix Table S3 has information related to all primers used.

## Northern and Western blot

Northern blotting of total cellular RNA was performed following the methods described previously (Ghosh *et al*, 2015). For miRNA detection, $^{32}$P-labelled 22 nt anti-sense LNA-modified probes specific for respective miRNAs or U6 snRNA were used. Densitometry of the blot was performed in Cyclone Plus Storage Phosphor System (Perkin Elmer), and Optiquant software (Perkin Elmer) was used for quantification.

Western analyses of different proteins were performed as described elsewhere. Detailed list of antibodies used for Western blot and immunoprecipitation is available as Appendix Table S4. Imaging of all Western blots was performed using an UVP BioImager 600 system equipped with VisionWorks Life Science software (UVP) V6.80 which was also used for quantification.

## Bio-informatics-based analysis of regulatory relationship between HuR and its target genes

An extensive list of HuR target genes (1,877) was compiled by utilizing the data from two previous analyses which had identified HuR targets either in HeLa cells or HEK293 cells by PAR-CLIP analysis (Lebedeva *et al*, 2011; Mukherjee *et al*, 2011). Target genes (1,103) that had binding site information, scored a log odds ratio (LOD) $\geq 0$ [where LOD is indicative of probability of an mRNA being associated with HuR], were differentially expressed either at protein/mRNA level upon HuR knock-down in HEK293 cells were compiled from Mukherjee *et al* (2011). Similarly, another set of target genes (875) identified based on PAR-CLIP analysis that showed differential expression either at mRNA/protein level upon HuR knock-down in HeLa cells were obtained from Lebedeva *et al* (2011). Further, based on the assumption that HuR may have similar target genes in macrophages, differentially expressed HuR target genes in isolated *Ld*-infected human macrophages (16 h) [fold change: 1.5, *P*-value: 0.05] were determined considering the GEO dataset GSE360 (Chaussabel *et al*, 2003). Additionally, RNAseq profile of mRNA from HuR (*Elavl1*) knockout murine bone marrow-derived macrophages (Lu *et al*, 2014a) was determined considering the GEO dataset (GSE63199) (Afgan *et al*, 2018). Possible regulatory events that could be involved in this interaction between HuR and its target gene (PPP2CA) either directly at the mRNA level or via intermediates at the protein–protein interaction level (Szklarczyk *et al*, 2015), signalling level (Kanehisa & Goto, 2000) and miRNA level (Lebedeva *et al*, 2011) have been identified.

## Statistical analysis

All graphs and statistical significance were calculated using GraphPad Prism 5.00 (GraphPad, San Diego). Experiments were done for minimum of three times to get the *P* values. The sample size was chosen by convenience, and no exclusion criteria were used. All the animals and samples were used in the analyses. Subjective randomization was used. *In vitro* analyses were not blind but occasionally verified by independent researchers within the

## The paper explained

### Problem

Maintaining inflammatory homeostasis is an important event in immune cells. Macrophages, that act as first line defence against any invading pathogens, need to show balanced immune response to safeguard themselves from self-destruction at the time of pathogen killing. Parasites like *Leishmania* have developed clever strategy to evade the immune sentinels and continue to propagate within macrophages. Sodium stibogluconate (SSG)-resistant clinical isolates are of serious clinical concern as they shifted the immune balance of the infected macrophage more towards a strong anti-inflammatory one. How to stimulate the infected macrophage to a pro-inflammatory setting to clear the residing drug-resistant pathogens is a major challenge from the clinical point of view.

### Results

We have identified two major players HuR, a miRNA depressor protein, and PP2A, a phosphatase that favours the pro or anti-inflammatory responses, respectively. During *Ld* infection, the parasite reciprocally modulates HuR and PP2A which ensure parasite survival within the macrophage. Although HuR overexpression could successfully control the drug-sensitive strain of *Leishmania* (*Ld*$^S$), it was not sufficient alone to clear the resistant form (*Ld*$^R$). HuR restoration along with PP2A inhibition however effectively counters the strong anti-inflammatory response induced by *Ld*$^R$. Interestingly, both the factors work by influencing the phosphorylation of miRNA effector protein Ago2 to regulate miRNA activity against target cytokines.

### Impact

Our data suggest intricate regulation of inflammatory response by PP2A and HuR that works primarily by modulating Ago2 phosphorylation during infection and inflammation. This might hold true for any disease that involves alteration of inflammatory homeostasis. Therefore, HuR and PP2A would be new drug targets not only to treat inflammatory diseases but also to affect anti-inflammatory responses during invasion of host cells by pathogens like *Leishmania* or mycobacterium.

group. In all experiments, the variance was similar between groups. Individual *P*-values were calculated by using Student's *t*-test. Curve fitting, the mathematical modelling, was done with OriginLab data analysis and Graphing software. The animal selection and analysis process were blind as injection/numbering, and tissue collection was done independently.

## Ethics statement

BALB/c mice were obtained from CSIR-Indian Institute of Chemical Biology animal facility. Experiments were done according to the national regulatory guidelines stated by the Committee for the Purpose of Supervision of Experiments on Animals, Ministry of Environment and Forest, Govt of India. Prior permission related to mice experiments was obtained through Institutional Animal Ethics Committee.

**Expanded View** for this article is available online.

## Acknowledgements

We acknowledge Witold Filipowicz and Gunter Meister for different plasmids constructs. SNB is supported by The Swarnajayanti Fellowship (DST/SJF/LSA-

03/2014-15) from Dept. of Science and Technology, Govt. of India, while AG, KM, AM, SG and IM received their support either from CSIR, India, University Grant Commission (UGC) or Dept. of Science and Technology, Govt. of India. SR is supported by J C Bose Fellowship, DST, Govt. of India (SB/S2/JCB-65/2014). The work also received support from a Indo-Swiss Bilateral Project Grant from Department of Biotechnology, Govt of India (BT/IN/Swiss/53/SNB/2018-19) and a High Risk High Reward Grant (HRR/2016/000093) from Dept. of Science and Technology, Govt. of India.

## Author contributions

AG, KM, AM and SG done all the experiments. Appendix Fig S1 and part of Fig 6 done by IM and KC. SNB conceptualized the idea, analysed data and wrote the manuscript along with AG, KM, AM and SG. KM also contributed in idea conceptualization and manuscript language editing. SC, SR and SS helped in planning some of the experiments described in Figs 6 and 8.

## Conflict of interest

The authors declare that they have no conflict of interest.

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
