## [Review Process File · EMBO Molecular Medicine]

MicroRNA Exporter HuR Clears the Internalized Pathogens by Promoting Pro-inflammatory Response in Infected Macrophages

Avijit Goswami, Kamalika Mukherjee, Anup Mazumder, Satarupa Ganguly, Ishita Mukherjee, Saikat Chakrabarti, Syamal Roy, Shyam Sundar, Krishnananda Chattopadhyay and Suvendra N. Bhattacharyya

Review timeline:

Submission date:	16 June 2019
Editorial Decision:	25 July 2019
Revision received:	30 November 2019
Editorial Decision:	19 December 2019
Revision received:	24 December 2019
Accepted:	8 January 2020

Editor: Céline Carret

Transaction Report:

1st Editorial Decision

25 July 2019

Thank you for the submission of your manuscript to EMBO Molecular Medicine. We have now heard back from the three referees whom we asked to evaluate your manuscript.

You will see from the set of comments pasted below, that overall, the referees find the study of interest. Still, they do have some reservations and suggestions to make the paper stronger. Referees 1 and 2 request *in vivo* data to validate the main *in vitro* findings, while referee 3 would like to see more parasite work and mechanism. All referees also commented on missing details, statistical analyses, discussions and explanations. Upon our cross-commenting exercise, referees agreed that a full analysis using different species and strains of *Leishmania* would certainly be interesting (as suggested by Ref. 3) but maybe not necessary for this article. Indeed, different species of *Leishmania* cause different disease outcome and the miRNA mediated mechanism could therefore be different in a different *Leishmania* species. Still, KO parasites being available, we would like to encourage you to test the *L. major* GP63-KO and WT strains in your setting as we believe that such an experiment would strengthen the mechanistic insights and novelty of the paper.

We would therefore welcome the submission of a revised version within three months for further consideration and would like to encourage you to address all the criticisms raised as suggested to improve conclusiveness and clarity. Please note that EMBO Molecular Medicine strongly supports a single round of revision and that, as acceptance or rejection of the manuscript will depend on another round of review, your responses should be as complete as possible. EMBO Molecular Medicine has a "scooping protection" policy, whereby similar findings that are published by others during review or revision are not a criterion for rejection. Should you decide to submit a revised version, I do ask that you get in touch after three months if you have not completed it, to update us on the status.

I look forward to receiving your revised manuscript.

***** Reviewer's comments *****

Referee #1 (Remarks for Author):

This paper assesses how *Leishmania donovani* regulates TNF- α , IL-6, and IL-1 in macrophages through regulation of miRNAs. They demonstrate that *L. donovani* utilizes a combination of upregulating Protein Phosphatase 2A that acts on Ago2 to keep it associated with miRNAs and downregulating the miRNA derepressor protein HuR. The authors completed a complex set of biochemical experiments to sort out a complicated regulation pathway that was no less than heroic. While the results are important for the field, the manuscript should be improved so that readers can better follow the experimental procedures and results. Addressing the following points will help improve the manuscript.

1. While generally well written, the manuscript would benefit from editing throughout the manuscript, particularly to correct many instances where the subject and verb are not in agreement (singular/plural). For example in the abstract the sentence "...the pathogen targets HuR to promotes" should be "to promote".
2. Throughout the manuscript broad statements about miRNAs and proinflammatory cytokines are used. Unless all miRNAs and cytokines were assessed, the authors should only state the specific miRNAs or cytokines that were assessed.
 - a. For example on page 8 (first sentence of paragraph 2), "...we documented restoration of miRNA activity...", should be "...we documented restoration of let-7a activity..".
 - b. Another example in the same paragraph, "...levels of miRNA regulated cytokine mRNAs were also dropped", state the specific cytokines assessed.
 - c. Figure 5C, state the specific miRNA being assessed in the figure and the text.
3. Methods.
 - a. In general, the methods state that the experiments were done a minimum of 3 times. Giving the n value for each figure would be more appropriate. For the qRT-PCR analysis, how many technical repetitions were performed and how many biological replicates?
 - b. Need to state how the relative levels of mRNA was calculated. Was it with the deltadelta CT method?
 - c. The western blot and densitometry methods need to be provided.
 - d. For infections, what phase were the parasites (e.g. log or stationary).
 - e. Parasite infections are listed for RAW cells. Was the infection for primary macrophages different? If not, heading should be changed to "Parasite infection of macrophages".
4. Figure 1E. Heavy chain is used as a control. More description of this protein should be given. Heavy chain of what?
5. For many of the assays the authors combine ectopic expression of FLAG-HA-Ago2 with immunoprecipitation and western blot analysis. The text, figures and legends should explicitly state what antibody was used to immunoprecipitate and what antibody was used to detect.
 - a. Keep the nomenclature the same throughout the manuscript. It seems that FH-Ago2 would be best because it indicates both the FLAG and the HA tag.
 - b. Figure 1E states that the IP antibody was anti-FLAG and the Western antibody was anti-HA. The legend seems to indicate that the IP was done with the HA antibody, "...band intensities were normalized against total Ago2 detected with HA specific antibody that was also used for Ago2 pull down."
 - c. Figure 1J legend doesn't state what antibody was used for the IP.
6. Figure 1I. It looks like siMKP3 and siSHP1/2 upregulated Ago2-associated let-7a. Was this statistically significant? If so, a sentence acknowledging this result and some discussion on what it might mean is warranted.
7. Figure 1K. It seems strange that the mir-146a result wasn't significant. Was this because that particular condition wasn't repeated as many times (see point #2 above).
8. Figure 2A could benefit from some densitometry analysis.

9. Figure 2B, label that the numbers at the bottom are "relative expression".
10. Figure 2F y axis should be labelled with the specific miRNA being assessed.
11. Figure 3F. Define BC in the legend.
12. Figure 3F. It appears that there might be lower levels of Ago2 phosphorylation with the SHP1 and MKP3 conditions. This figure could benefit by statistical analysis of the phosphorylation levels from more than one experiment.
13. Figure 3H. Explain in the legend what the asterisks and the 'ns' is compared to in the legend.
14. Figure 4A Label the Y axis with mRNA
15. Figure 4A. Were the analyses for the primary macrophages not significant? If so, this should be mentioned in the text or maybe additional experiments could be performed to reach significance.
16. Figure 4C. Please report infections as number of parasites per 100 cells or also provide the % of cells infected AND number of parasites/infected cell.
17. Figure 4D. What happens if the unit comparison was set at the uninfected, siCon, WT condition for all of the conditions? How different are the IL-10 mRNA levels between the WT FH-Ago2 and the FH-Ago2 Y529Y treatments? Maybe this is what is presented in Figure 4F?
18. Figure 4F. Are these cells Ago2 depleted?
19. Figure 4E. Show data of parasite loads between WT and mutants as it says in the legend. Also show the parasite loads for the -OA and +OA.
20. The first sentence in the legend of Figure 5 states "...export from mammalian liver cells." Were these experiments performed in liver cells? Figure 5A indicates RAW264.7 cells.
21. The legend says 24 hours of treatment, but Figure 5A says 4 hours. Which is it?
22. Figure 5D. Is the concentration really particles/ml? Please check units.
23. Figure 5E, the last sentence of the legend is copied twice.
24. Should state in the text if mir-155 has been shown to target TNF, IL-6 and/or IL-1.
25. Figure 5I. For this figure the unit of comparison should be PCI-Neo-LPS for all conditions. The way the data is presented now, it looks like LPS doesn't upregulate TNF.
26. Figure 5K. As most experiments have a 24 hour time point, HuR expression should be assessed up to that time point.
27. Figure 5M does not provide convincing data that the HuR is primarily coming from the ER and polysome. Perhaps some densitometry would make the result clearer.
28. The GP63 cleavage of HuR is extremely interesting. It would be good to test in vivo using a GP63 KO line.
29. Figure 6A-C figure legend uses the name 'ELAV1'. It would be less confusing to use 'HuR' because that is what is primarily used in the text. Or put HuR in parentheses.
30. Figure 7D. Is each lane a separate mouse? State what each lane is in the legend or in the figure.
31. Figure 7G. Presenting CT values is not intuitive. Although a higher CT value indicates lower parasite numbers, the reader really has to think about it. It would be better to show the relative parasite numbers. What is Ld count, maybe it is a parasite burden calculated from the CT values. If this is the case then the figure is alright. However, the legend needs to state this explicitly and how the LD count was calculated.
32. Figure 7H. See point #31 above. Again reporting CT values is not intuitive.
33. Figure 7I. Why is LdS done for 12hr and LdR for 4 hours. The timepoints should be the same.
34. Figure 7J. Was any of the data statistically significant? Were these technical reps or biological reps? See point #31 above.

Referee #2 (Comments on Novelty/Model System for Author):

These experiments need to be tested in in vivo models. Please see my comments below.

Referee #2 (Remarks for Author):

The manuscript by Goswami et al., describes the tripartite regulatory axis involving PP2A, Ago2 and HuR proteins in the regulation of pro- and anti-inflammatory cytokine response in following Leishmania infection of macrophages. The study attempts to advance the understanding of the role of microRNA mediated immune regulation in parasitic infections. Overall the study builds on the previous work done by the authors. However, the manuscript needs language revision to improve readability. At several places in the manuscript the authors make out of context statements or fail to provide sufficient background references. The discussion section does not do complete justice to the breadth of the findings and the overarching impact of the results described in the manuscript.

Major comments:

1. Fig. 1E: Differences in Relative values are small. How many times these experiments were performed?
 2. Fig. 1F: Please show the Relative values between different groups.
 3. Fig. 1H: Is the difference with SiPP2A treatment significantly different, since similar differences in other groups in not?
 4. Fig. 1K: Is the difference in Ago2 bound mRNA with miRNA 146 significant in SiPP2A treatment?
- Overall the Fig. 1 is very crowded and there is sloppiness in defining the significance.
5. Fig. 2D: The in vitro phosphatase assay the numbers do not appear to accurately represent the density of pYago2 product. Perhaps this is the limitation of the assay. Since similar point is made in Fig 2B, Fig 2D does not add a whole lot of value.
 6. Fig. 3C: There is no data showing increased anti-inflammatory or decrease in pro-inflammatory responses in primary macrophages as mentioned in the text on page 11 line 18.
 7. Fig. 3K and M: No mention of the data in these figures in the text.
 8. Fig. 4A: Are the differences in IL-10 and IL-6 levels significant in primary macrophages upon treatment with Okadiac acid (OA)?
 9. The manuscript almost entirely focuses on TNF, IL-6 and IL-1b. The authors show that depletion of PP2A resulted in downregulation of IL-10 production. They also show that IL-10 production is determined by the phosphorylation-dephosphorylation of Ago2. This is not pursued further in the later experiments involving HuR and EV studies.
 10. The authors cite TNF α mRNA to be a target for HuR binding that allows it to peel off miRNA and transport through EVs. While this is consistent with the observations in the current manuscript, what mechanisms might be involved in enabling the transcription of IL-6, IL-1b and possibly IL-10 mRNAs remains unexplained.
 11. The HuR expression peaked at 4h post-infection with *L. donovani* parasites (Fig 5K). At 6h post infection, there was no detectable HuR expression. However the TNF and IL-6 expression peaked at 3hr of LPS treatment. Showing whether the kinetics of TNF, IL-6 expression matches the HuR expression pattern would add weight to the conclusions. Also what happens to the expression of HuR at later time points of *L. donovani* infection? Does this data mean the role of HuR is limited to the early time points of infection only?
 12. The authors seem to suggest that HuR mediated transporting of miRNAs out of the cell effectively takes them out of action. This is inconsistent with the emerging literature on the broad immunogenic potential of miRNA laden EVs in microbial infections and anti-cancer therapies (see recent studies from Breakfield XO, Regev-Rudzki, Jeff Schorey, Raghu Kalluri labs).
 13. In the bioinformatic analysis, the reasoning provided to include all potential targets based on the assumption that HuR may have similar targets genes in macrophages and cells of neuronal origin does not appear sound. Do the authors find the same targets if they limit their search to macrophages and immune cells rather than all kinds of cells in their analysis?
 14. On page 20, first paragraph last few lines: the authors mention "However reciprocal actions of HuR and PP2A in macrophages.....to cure infection by LdR". To back this statement authors need to demonstrate the effects observed in vitro can be replicated in in vivo. Therefore the major point missing in this study demonstration of the role of PP2A and HuR in regulating inflammatory response in macrophages via miRNA in vivo using knock out mice.

Minor comments:

1. The manuscript needs a thorough language revision. There are several imprecise statements in the manuscript that need to be corrected. A sampling of the errors is listed below.
2. In the introduction the authors state "Leishmania not only stops the antibody production in the host.....This is not an accurate statement.
3. Reduction of cytokine mRNA levels after prolonged exposure of macrophage to LPS has been noted and it has been anticipated as... Please cite the reference for this observation.
4. The proteins in stressed human hepatocytes... NOT protein in stressed human hepatocytes.
5. Page 11: Ago2 phosphorylation happens primarily during LPS or PMA mediated activation of macrophages... Where is this observation coming from? Prior studies?
6. Page 12: Define 'MVB associated fraction'
7. Page 13: It was accompanied by a decrease...should be 'by a decrease.'
8. Page 16. Are the numbers in parenthesis referring to citations? Certain targets (62) and... certain HuR targets (20)... or some kind of target counts?

Referee #3 (Comments on Novelty/Model System for Author):

This manuscript addresses the role of a macrophage pathway in the modulation of pro-inflammatory response upon exposure to LPS and *Leishmania donovani*. In this regard, there is little medical impact in a previsible future.

The model system could be better: in the field of *Leishmania* research, the use of primary cells is always preferable over the use of macrophage cell lines.

Referee #3 (Remarks for Author):

Goswami and colleagues have investigated the role and regulation of HuR in the control of the inflammatory response of macrophages exposed to either LPS or the protozoan parasite *Leishmania donovani*. This manuscript contains novel information on the host cell pathways targeted by *Leishmania* to down-modulate the expression of pro-inflammatory genes. However, a number of issues remain to be addressed to strengthen the conclusions of this study.

1. It is unclear whether the pathway described for *L. donovani* also works for other *Leishmania* species known to induce an inflammatory response, such as *L. major* or *L. mexicana*. Taking advantage of these differences among *Leishmania* species would strengthen the conclusions of this manuscript. Related to this, the fact that another strain of *L. donovani* used in this study (the Sb-resistant strain BHU569), which was shown to induce higher levels of IL-10 than the AG83 strain, also stimulates the expression of higher levels of PP2A raises the issue of mechanism. The authors refer to the Sb-S and Sb-R strains, but offer little explanations for the differential ability to stimulate IL-10 or PP2A expression. Is it related to the expression of a *Leishmania* gene related resistance to antimony? What is the molecular basis for antimony resistance in that strain?

2. One major weakness of this manuscript is the absence of mechanism to explain how *Leishmania* alters the host cell processes under investigation. For instance, what triggers PP2A expression in cells exposed to *Leishmania donovani*? Is it the phagocytic process? This can be addressed by including inert particles as a phagocytosis control. The possibility that the surface glycolipid LPG is involved in this process also deserves to be investigated, since this *Leishmania* virulence factor activates TLR4. The role of LPG could be addressed using LPG mutants for *L. donovani*. Similarly, the authors provide indirect evidence suggesting that the metalloprotease GP63 cleaves HuR in infected macrophages. Given the availability of the *L. major* GP63-deficient mutant, the authors could take the opportunity to directly assess the role of GP63 in HuR.

3. For Fig 4A, at the bottom of page 12, the authors state that "this was accompanied by increased production of pro-inflammatory cytokines". This is a misleading statement, as the authors have measured mRNA levels by PCR. In this regards, the information related to the measurement of cytokine mRNAs is missing in the Materials and Methods section. Additionally, on Fig 4A, "Relative level IL-10 or IL-6" has to be changed to reflect the reality. The same applies to Fig 5I. In fact, actual cytokine levels should be measured.

4. In Fig 4B, C, E, quantification of phagocytosis is necessary. In addition, the authors must assess parasite burden at 2h and 24 h post-phagocytosis in order to clarify whether PP2A is involved in phagocytosis per se, or controls a process associated to the microbicidal function of macrophages.

5. Fig 7F and G, in addition to the liver, the authors should analyze what is going on in the spleen in terms of parasite burden and cytokine levels. It would also be appropriate to show the expression of HA-HuR in the organs of mice injected with the Ha-HR expression vector.

1st Revision - authors' response

30 November 2019

Referee #1 (Remarks for Author):

This paper assesses how Leishmania donovani regulates TNF- α , IL-6, and IL-1 β in macrophages through regulation of miRNAs. They demonstrate that L. donovani utilizes a combination of

upregulating Protein Phosphatase 2A that acts on Ago2 to keep it associated with miRNAs and downregulating the miRNA derepressor protein HuR. The authors completed a complex set of biochemical experiments to sort out a complicated regulation pathway that was no less than heroic. While the results are important for the field, the manuscript should be improved so that readers can better follow the experimental procedures and results. Addressing the following points will help improve the manuscript.

1. While generally well written, the manuscript would benefit from editing throughout the manuscript, particularly to correct many instances where the subject and verb are not in agreement (singular/plural). For example in the abstract the sentence "...the pathogen targets HuR to promotes" should be "to promote".

We apologize for the incorrect English and have now done a thorough language editing for the revised version to ensure that such grammatical mistakes can be avoided.

2. Throughout the manuscript broad statements about miRNAs and proinflammatory cytokines are used. Unless all miRNAs and cytokines were assessed, the authors should only state the specific miRNAs or cytokines that were assessed.

We very much agree with the reviewer's concern and have done necessary changes to mention specific cytokines and miRNAs that we have used in experiments.

a. For example on page 8 (first sentence of paragraph 2), "...we documented restoration of miRNA activity...", should be "...we documented restoration of let-7a activity..".

We have changed the phrase as recommended.

b. Another example in the same paragraph, "...levels of miRNA regulated cytokine mRNAs were also dropped", state the specific cytokines assessed.

We have now mentioned the names of the cytokines that we have followed.

c. Figure 5C, state the specific miRNA being assessed in the figure and the text.

We have edited the text and figures

3. Methods.

a. In general, the methods state that the experiments were done a minimum of 3 times. Giving the n value for each figure would be more appropriate. For the qRT-PCR analysis, how many technical repetitions were performed and how many biological replicates?

The exact number of experimental set that the data represents in individual cases now has been mentioned against each panel/figures. The exact P values are listed in Appendix Table 2 for each panel.

b. Need to state how the relative levels of mRNA was calculated. Was it with the deltadelta CT method?

Yes, the relative fold change was calculated using the delta delta CT method and has been clearly mentioned in the methods section now.

c. The western blot and densitometry methods need to be provided.

We are sorry for not providing the details of the methods and have now them included them in the text methods section.

d. For infections, what phase were the parasites (e.g. log or stationary).

Stationary phase parasites were used for every experiment. Also, parasites in between 2nd to 5th passage were used. This has now been clearly indicated in the text method part.

e. Parasite infections are listed for RAW cells. Was the infection for primary macrophages different? If not, heading should be changed to "Parasite infection of macrophages".

The infection process does not differ significantly. However the difference in pre-infection condition for RAW and Primary cells (PEC) respectively has been mentioned in the methods part. The title of the section has also been changed as suggested.

4. Figure 1E. Heavy chain is used as a control. More description of this protein should be given. Heavy chain of what?

Yes that has been an unforced error and we have taken good care to include the details in figure legends of respective figures.

5. For many of the assays the authors combine ectopic expression of FLAG-HA-Ago2 with immunoprecipitation and westernblot analysis. The text, figures and legends should explicitly state what antibody was used to immunoprecipitate and what antibody was used to detect.

a. Keep the nomenclature the same throughout the manuscript. It seems that FH-Ago2 would be best because it indicates both the FLAG and the HA tag.

We accept the suggestions and have now used FH-Ago2 throughout the manuscript. We have also mentioned the details of antibodies used for detection and immunoprecipitation.

b. Figure 1E states that the IP antibody was anti-FLAG and the Western antibody was anti-HA. The legend seems to indicate that the IP was done with the HA antibody, "...band intensities were normalized against total Ago2 detected with HA specific antibody that was also used for Ago2 pull down."

We have now clarified the confusion and corrected the text of the corresponding legend section.

c. Figure 1J legend doesn't state what antibody was used for the IP.

We have mentioned the details of the antibody against Figure 1J .

6. Figure 1I. It looks like siMKP3 and siSHP1/2 upregulated Ago2-associated let-7a. Was this statistically significant? If so, a sentence acknowledging this result and some discussion on what it might mean is warranted.

Although the trend showed an increase in Ago2-associated let-7a level, however the values were found to be non-significant after the experiments were repeated for minimum 3 times, and we have mentioned this in the result part.

7. Figure 1K. It seems strange that the mir-146a result wasn't significant. Was this because that particular condition wasn't repeated as many times (see point #2 above).

miR-146a levels were found to be significant after four biological replicates included in the analysis. For IP experiments, biological replicates were preferred instead of technical replicates as the amount of RNA obtained was quite low to run technical replicates for each miRNA. Number of biological replicates used is mentioned in the respective legends.

8. Figure 2A could benefit from some densitometry analysis.

Densitometric analysis has been done and represented along with the western blot data.

9. Figure 2B, label that the numbers at the bottom are "relative expression".

Has been edited and incorporated as "relative amount".

10. Figure 2F y axis should be labelled with the specific miRNA being assessed.

It has now been mentioned.

11. Figure 3F. Define BC in the legend.

This has now been defined.

12. Figure 3F. It appears that there might be lower levels of Ago2 phosphorylation with the SHPI and MKP3 conditions. This figure could benefit by statistical analysis of the phosphorylation levels from more than one experiment.

Indeed, its a good suggestion and panel showing standard deviation at the phosphorylation levels has now been included in the figure.

13. Figure 3H Explain in the legend what the asterisks and the 'ns' is compared to in the legend.

Description has been incorporated in the legend in new version (Figure 3G).

14. Figure 4A Label the Y axis with mRNA

It has been done now.

15. Figure 4A. Were the analyses for the primary macrophages not significant? If so, this should be mentioned in the text or maybe additional experiments could be performed to reach significance.

Experiments were repeated and now value with statistical significance has been included.

16. Figure 4C. Please report infections as number of parasites per 100 cells or also provide the % of cells infected AND number of parasites/infected cell.

This is an important point mentioned by the reviewer. Data has been represented as number of parasites per 100 cells for Okadaic acid treatment as the effect of OA should be all cells present in the culture while it has been represented as number of parasites per infected cell in case of HA-HuR expression condition. This is important in the context of HA-HuR expression as we can expect to see the effect of HA-HuR on infection level observed for individual cell in a population where only the transfected cells would be expressing the HA-HuR. Therefore we want to keep the data presentation accordingly.

17. Figure 4D. What happens if the unit comparison was set at the uninfected, siCon, WT condition for all of the conditions? How different are the IL-10 mRNA levels between the WT FH-Ago2 and the FH-Ago2 Y529Y treatments? Maybe this is what is presented in Figure 4F?

Difference in IL-10 mRNA levels between the WT FH-Ago2 and the FH-Ago2Y529F expressing cells has been represented in Figure 4F. From that it is clear that FH-Ago2Y529F increases the IL-10 level. However the effect of infection on PP2A could have other implications as it could ensure IL-10 upregulation through a miRNA independent pathway. This possibility has been nullified in experiments done with siPP2A described in Figure 4D. The effect of PP2A depletion does not have a significant effect on IL-10 when cells are expressing a phosphorylation defective Ago2 mutant. In Figure 4D we preferred to take the value obtained with non-infected siCon treated cells as unit both for FH-Ago2 or FH-Ago2Y529F co-expression conditions. The representation otherwise may cause confusion as two variables like siRNA and FH-Ago2 (wild type and Y529F mutant) expression with or without *Leishmania* infection has been studied in experiments described in Figure 4D.

18. Figure 4F. Are these cells Ago2 depleted?

Yes, Ago2 was depleted using mouse specific siAgo2 while the FH-Ago2 or FH-Ago2Y529F constructs express human Ago2. We updated the information in the figure legend as well.

19. Figure 4E. Show data of parasite loads between WT and mutants as it says in the legend. Also show the parasite loads for the -OA and +OA.

Parasite load in terms of parasites per cells has been shown along with representative microscopic data. Data on parasite number for early and late phase of OA treatment has been shown in Figure 4G.

20. The first sentence in the legend of Figure 5 states "...export from mammalian liver cells." Were these experiments performed in liver cells? Figure 5A indicates RAW264.7 cells.

This was an incorrect statement and the necessary change has been done to correctly point out the experiments done with RAW264.7 cells.

21. The legend says 24 hours of treatment, but Figure 5A says 4 hours. Which is it?

It is 24h and the change has been done in Figure itself.

22. Figure 5D (Is it Fig 5D?). Is the concentration really particles/ml? Please check units.

Yes. The concentration is in the range of 10^9 particles/ml in Fig 5B

23. Figure 5E, the last sentence of the legend is copied twice.

The problem has been corrected.

24. Should state in the text if mir-155 has been shown to target TNF, IL-6 and/or IL-1.

From the literature miR-155 is not the validated regulator of these cytokines. However, miR-155 has been reported to target a TLR pathway adaptor protein Myd88 (Bandyopadhyay et al, 2014; Tang et al, 2010). Myd88 knockdown alters TNF- α , IL-1 β and IL-6 at various conditions suggesting a direct relation between miR-155 and cytokine production which is mediated by Myd88 (Lin et al, 2015; Morandini et al, 2013).

25. Figure 5I. For this figure the unit of comparison should be PCI-Neo-LPS for all conditions. The way the data is presented now, it looks like LPS doesn't upregulate TNF.

The data has been represented as suggested, also the absolute level of TNF- α protein is shown by ELISA method as also suggested by reviewer 3.

26. Figure 5K. As most experiments have a 24 hour time point, HuR expression should be assessed up to that time point.

Blot has been replaced with blots showing HuR expression upto 24hours in both RAW 264.7 cells as well as primary macrophage cells. Two other Leishmania species as suggested by Reviewer 3 have been included in the analysis.

27. Figure 5M does not provide convincing data that the HuR is primarily coming from the ER and polysome. Perhaps some densitometry would make the result clearer.

We performed the densitometry and have included it in the main figure as separate panel after Figure 5M.

28. The GP63 cleavage of HuR is extremely interesting. It would be good to test in vivo using a GP63 KO line.

This is a very important point that has also been raised by Reviewer 3. We have done additional experiments to support our claim on GP63 –mediated HuR cleavage. On that line, we reconstituted liposomes with purified GP63 and with that we have been able to show that HuR gets cleaved in cells treated with GP63 containing liposomes (EV3 F-I). Similar experiments done *in vivo* also gave us similar results where GP63 containing liposomes lowered HuR level in liver tissue similar to what we have observed in *Ld* infected cells (Figure 7E). This is rather a more convincing way of showing that GP63 is the protein that cleaves HuR during *Ld* infection. We have done similar

experiments in one of our previous manuscript to show GP63 also targeting Dicer1 in liver cells (Ghosh et al, 2013).

GP63 KO strain for *Ld* is not available and *Ld* has been primarily used throughout this study to get majority of the data. The *L. major* GP63 KO strain could be a choice but considering the defined role of GP63 in internalization of the parasite in target macrophage, the conclusion of experiments with GP63 KO strain would never be conclusive (Brittingham et al, 1999). The expected reduced effect on HuR cleavage with GP63 KO strain could either be due to defective internalization of the parasite in absence of GP63 on interacting parasite surface or it could also be due to GP63's direct effect on HuR cleavage step after the entry. Arguably HuR cleavage could be done by other proteases of the parasite origin whose entry would have been restricted to target cells due to lack of GP63 and its effect on parasite internalization. Therefore, with GP63 KO cells, it would never be convincing to claim that GP63 is the candidate protease that cleave HuR in infected cells and thus solely responsible for the effect on HuR. With GP63 reconstituted liposomes, we could effectively show that GP63 is solely responsible for HuR degradation observed in cells infected with GP63 containing parasites.

29. Figure 6A-C figure legend uses the name 'ELAV1'. It would be less confusing to use 'HuR' because that is what is primarily used in the text. Or put HuR in parentheses.

We have changed the legends as suggested.

30. Figure 7D. Is each lane a separate mouse? State what each lane is in the legend or in the figure.

Yes, each lane stands for separate mouse. Corrections have been made both in the legend and figure.

31. Figure 7G. Presenting CT values is not intuitive. Although a higher CT value indicates lower parasite numbers, the reader really has to think about it. It would be better to show the relative parasite numbers. What is Ld count, maybe it is a parasite burden calculated from the CT values. If this is the case then the figure is alright. However, the legend needs to state this explicitly and how the LD count was calculated.

Ct value plots have been supplemented with relative fold change plots. *Ld* count is the parasite burden calculated from the Ct values using *Ld* specific primers. We have taken good care to describe that in details in the figure legends.

32. Figure 7H. See point #31 above. Again reporting CT values is not intuitive.

Changes have been made as suggested. Ct value plots have been supplemented with relative fold change plot.

33. Figure 7I. Why is LdS done for 12hr and LdR for 4 hours. The timepoints should be the same.

That was an error in data presentation as we have done all the experiments with Ld^S and Ld^R for same time duration. It has now been corrected in respective figure panel (new Figure 8E).

34. Figure 7J. Was any of the data statistically significant? Were these technical reps or biological reps? See point #31 above.

The data has been replaced with data showing statistical significance having more replicates (New Figure 8F).

Referee #2 (Comments on Novelty/Model System for Author):

These experiments need to be tested in in vivo models. Please see my comments below.

We have largely addressed this concern in additional experiments done in animal models. Please see our reply below.

Referee #2 (Remarks for Author):

The manuscript by Goswami et al., describes the tripartite regulatory axis involving PP2A, Ago2

and HuR proteins in the regulation of pro- and anti-inflammatory cytokine response in following Leishmania infection of macrophages. The study attempts to advance the understanding of the role of microRNA mediated immune regulation in parasitic infections. Overall the study builds on the previous work done by the authors. However, the manuscript needs language revision to improve readability. At several places in the manuscript the authors make out of context statements or fail to provide sufficient background references. The discussion section does not do complete justice to the breadth of the findings and the overarching impact of the results described in the manuscript.

We acknowledge the points made by the reviewer and we have rewritten few parts to "do justice to the breadth of the findings and overarching impact of the results". This has been an encouraging statement from the reviewer and we like to thank him for that.

We have done substantial language editing as well to improve the flow and readability of the manuscript.

Major comments:

1.Fig. 1E: Differences in Relative values are small. How many times these experiments were performed?

The experiments have been repeated for multiple times and the relative band intensities with standard deviation have now been plotted.

2.Fig. 1F: Please show the Relative values between different groups.

We have plotted the relative intensity values but now it is a part of EV1 A-B.

3.Fig. 1H: Is the difference with SiPP2A treatment significantly different, since similar differences in other groups in not?

Yes, the difference with siPP2A treatment is significantly different having less standard deviation.

4.Fig. 1K: Is the difference in Ago2 bound mRNA with miRNA 146 significant in SiPP2A treatment?

Yes, the data is significant. New graph has been drawn after including additional replicates of experiments and now has been used to replace the previous graph.

Overall the Fig. 1 is very crowded and there is sloppiness in defining the significance.

Unfortunately there has been hardly any option left rather than moving a part of the main data to Extended View Figure1 (EV1) and that we have done now by moving old panel 1F from previous version to EV1 as panel A. Care has been taken to include data with significance as appropriate.

5.Fig. 2D: The in vitro phosphatase assay the numbers do not appear to accurately represent the density of pYAgo2 product. Perhaps this is the limitation of the assay. Since similar point is made in Fig 2B, Fig 2D does not add a whole lot of value.

Fig 2D is important as it involves use of a mutant form of Ago2 (Y529F) that is phosphorylation defective at 529 position and thus should not get influenced by PP2A. Fig 2D signifies that PP2A can dephosphorylate WT Ago2 *in vitro*, however this was not happening with Y529F Ago2. This data suggests that PP2A mediated dephosphorylation is effective in removing the phosphate from Y529, the Tyr residue known to be important for miRNA binding of Ago2 (Rudel & Meister, 2008). Hence, the data suggest that PP2A mediated dephosphorylation specifically occurs for the Y529 residue. The intensity were measured thrice to get the value and then normalized against Ago2 band intensity. The apparent inaccuracy may be contributed by difference in Ago2 band intensities between samples!

6.Fig. 3C: There is no data showing increased anti-inflammatory or decrease in pro-inflammatory responses in primary macrophages as mentioned in the text on page 11 line18.

Fig 3C lower panel shows primary macrophage (PEC) data showing the mRNA levels of IL-10 (anti-inflammatory) and IL-6 and TNF- α (pro-inflammatory).

7.Fig. 3K and M: No mention of the data in these figures in the text.

We apologies for the same and have incorporated the description in the relevant portion of the revised manuscript result section (New Figure 3L and N).

8.Fig. 4A: Are the differences in IL-10 and IL-6 levels significant in primary macrophages upon treatment with Okadiac acid (OA)?

Yes, the difference in the levels is now significant and has been shown in the respective graphs.

9.The manuscript almost entirely focuses on TNF, IL-6 and IL-1b. The authors show that depletion of PP2A resulted in downregulation of IL-10 production. They also show that IL-10 production is determined by the phosphorylation-dephosphorylation of Ago2. This is not pursued further in the later experiments involving HuR and EV studies.

This is an interesting point. We have detected strong induction of pro-inflammatory response upon HuR expression. We have tried to measure the change in the levels of IL-10. However, as IL-10 mRNA level is very low in naive cells they were undetectable in HuR expressing cells where it is expected to get further downregulated! These technical problems did not allow us to include the data on IL-10 in the HuR expression or EV contexts, where primary effect of HuR on pro-inflammatory pathway seems to be very prominent with upregulated TNF- α and IL-1 β expression.

10.The authors cite TNFa mRNA to be a target for HuR binding that allows it to peel off miRNA and transport through EVs. While this is consistent with the observations in the current manuscript, what mechanisms might be involved in enabling the transcription of IL-6, IL-1b and possibly IL-10 mRNAs remains unexplained.

Interesting work by other group has shown previously that HuR regulates TNF- α induced IL-6 production by stabilizing IL-6 mRNA that otherwise gets negatively regulated by another RNA binding protein TTP (Shi et al, 2012). Consistent with this finding, other groups have reported LPS induced p38 pathway mediates phosphorylation of TTP which alters its inhibitory effect on IL-1 β mRNA levels (Chen et al, 2006; Neuder et al, 2009). IL-1 β also has binding sites for HuR thus HuR directly also can influence its expression. IL-10 mRNA expression was undetectable in HA-HuR expressing cells and thus we could not comment on that.

11.The HuR expression peaked at 4h post-infection with L donovani parasites (Fig 5K). At 6h post infection, there was no detectable HuR expression. However the TNF and IL-6 expression peaked at 3hr of LPS treatment. Showing whether the kinetics of TNF, IL-6 expression matches the HuR expression pattern would add weight to the conclusions. Also what happens to the expression of HuR at later time points of L donovani infection? Does this data mean the role of HuR is limited to the early time points of infection only?

HuR expression gradually decrease with Ld infection as depicted in the new 5K panel where later time points have been included. In LPS treated cells expression of HuR increases with time (Figure 6D). The kinetics of IL-6 expression has already been followed against HuR expression in the same panel. The TNF- α also followed the same pattern as shown in Figure 1C.

The later time points of Ld infection has now been included in 5K to show the HuR levels in late infected cells and is consistent with low HuR detected in animal liver after infection where a steady state condition should prevail when the decreased expression of HuR was detected.

12.The authors seem to suggest that HuR mediated transporting of miRNAs out of the cell effectively takes them out of action. This is inconsistent with the emerging literature on the broad immunogenic potential of miRNA laden EVs in microbial infections and anti-cancer therapies (see recent studies from Breakfield XO, Regev-Rudzki, Jeff Schorey, Raghu Kalluri labs).

The immunogenic properties of the EV are dependent on the cargos they carry. We have enough evidence to show, from our unpublished results, the immunostimulatory role of liver specific miR-

122 that can be transferred via EVs to neighbouring macrophage cells to get them stimulated (Saha et al, 2018; Xu et al, 2018). Similar to what has been described in our previous paper where role of HuR in enhancing EV-mediated export of miR-122 has been described, we can attribute a role of HuR in contributing the inflammatory response propagation in that context (Basu & Bhattacharyya, 2014; Mukherjee et al, 2016).

Therefore the effect could be very much contextual. In macrophage, HuR by exporting out repressive let-7a or similar miRNAs during early phase of LPS- stimulation allows pro-inflammatory cytokines to get expressed and thus ensures pro-inflammatory response to set in HuR expressing cells. Interestingly miR-155, a pro-inflammatory mRNA (Squadrito et al, 2013) also get exported out from macrophage expressing HuR. This pro-inflammatory signal is communicated to neighbouring naive macrophages to get them stimulated via a secondary pathway. Therefore, the data is not inconsistent with previously published data rather in line with the findings by us and others on both pro- and anti-inflammatory role of EVs. It could be determined by the cargos they carry!

13. In the bioinformatic analysis, the reasoning provided to include all potential targets based on the assumption that HuR may have similar targets genes in macrophages and cells of neuronal origin does not appear sound. Do the authors find the same targets if they limit their search to macrophages and immune cells rather than all kinds of cells in their analysis?

HuR is primarily known to associate with specific upstream/downstream sequences of target mRNAs, for instance, U/AU-rich sequences either at 3'-untranslated regions (UTRs) or within pre-mRNA introns (Srikantan & Gorospe, 2011). Since RNA-HuR interactions have been extensively characterized in PAR-CLIP analysis considering HeLa or HEK293 cells in previous works (Lebedeva et al, 2011; Mukherjee et al, 2011), we have utilized the data provided to generate a list of HuR target mRNAs that may be regulated also in macrophages. Subsequently, by determining whether any of these probable HuR targets are differentially expressed when human macrophages are infected with *Ld*, we could predict 146 likely mRNA targets of HuR that may have a role in macrophages in *Ld* infection scenario. During *Ld* infection, HuR expression goes down. Hence, the underlying assumption was that as the binding sites of the target genes would be present in the macrophage RNA as well; HuR mediated regulation of target genes' expression would also be altered during *Ld* infection. Additionally, considering RNAseq profile of mRNA from HuR (*Elavl1*) knockout murine bone marrow-derived macrophages (Lu et al, 2014) described in the GEO dataset (GSE63199), we could observe that 136 among the previously predicted 146 targets show differences in their expression profile under these conditions.

14. On page 20, first paragraph last few lines: the authors mention " However reciprocal actions of HuR and PP2A in macrophages.....to cure infection by LdR". To back this statement authors need to demonstrate the effects observed in vitro can be replicated in in vivo. Therefore the major point missing in this study demonstration of the role of PP2A and HuR in regulating inflammatory response in macrophages via miRNA in vivo using knock out mice.

Yes! That has been an interesting point as well as a very difficult point to prove in animal experimental context. We have designed a separate set of experiments now in the revised version to include where animals, treated in combination or alone with PP2A inhibitor and HA-HuR expression plasmid, were challenged with drug resistant form of the parasite. Consistent with the cell line data (Figure 8F and EV4), the strong inflammatory response along with an effect on parasite number were detected only when HA-HuR was expressed and PP2A was inhibited in mouse model (Figure 8 G and H)! Thus this data has now allowed us to be in a more comfortable position to claim that the reciprocal action of HuR and PP2A in controlling the balance of inflammatory response can be tapped to cure infection by the drug resistant form of the Leishmania pathogen.

Minor comments:

1. The manuscript needs a thorough language revision. There are several imprecise statements in the manuscript that need to be corrected. A sampling of the errors is listed below.

Thank you for pointing the limitation and we apologize for the mistakes in language and choice of words. We have now taken precaution and thoroughly checked the manuscript for its accuracy in language and presentation.

2. In the introduction the authors state "Leishmania not only stops the antibody production in the host.... This is not an accurate statement.

We have changed that.

3. Reduction of cytokine mRNA levels after prolonged exposure of macrophage to LPS has been noted and it has been anticipated as ... Please, cite the reference for this observation.

Reference has been included

4. The proteins in stressed human hepatocytes...NOT protein in stressed human hepatocytes.

This has been corrected.

5. Page 11: Ago2 phosphorylation happens primarily during LPS or PMA mediated activation of macrophages... Where is this observation coming from? Prior studies?

Yes. The reference now has been cited.

6. Page 12: Define 'MVB associated fraction'

We have included the details and explanation

7. Page 13: It was accompanied by a decrease...should be 'by a decrease.'

We have corrected this.

8. Page 16. Are the numbers in parenthesis referring to citations? Certain targets (62) and... certain HuR targets (20) ... or some kind of target counts?

These are number of targets in those categories.

Referee #3 (Comments on Novelty/Model System for Author):

This manuscript addresses the role of a macrophage pathway in the modulation of pro-inflammatory response upon exposure to LPS and Leishmania donovani. In this regard, there is little medical impact in a previsible future.

Although the manuscript describes the findings in the context of *Ld* infection and LPS treatment, the implications of these findings are much bigger than that and thus we like to have a different view refer to the Reviewer's view on medical impact of the study.

Balancing of inflammatory vs. anti-inflammatory responses in mammalian system is not only important in the context of infection by a pathogen, but also it has immediate consequence in other systemic and chronic diseases we may think off. For example the inflammatory response is very much linked with tumour microenvironment and cancer. Thus a systematic understanding of the mechanism that governs the pattern formation within inflammatory response circuit in higher mammals should be considered to be a very important step forward and towards identification of new classes of immunomodulators. These modulators, specifically targeting important candidate molecule, can be used for curtailing infectious and non-infectious diseases. HuR and PP2A are the two key players that we identified in balancing this action. Therefore, the work should be considered to be a timely concept development that has been extensively tested and established through the work described in this manuscript in an infection/host-pathogen interaction context.

The model system could be better: in the field of Leishmania research, the use of primary cells is always preferable over the use of macrophage cell lines.

Following this suggestions we have now done extensive *in vivo* experiments and *ex vivo* primary cell based data to strengthen our claim. All the major inferred conclusions are now been verified or substantiated both in primary cells and in animal model of infection.

Referee #3 (Remarks for Author):

Goswami and colleagues have investigated the role and regulation of HuR in the control of the inflammatory response of macrophages exposed to either LPS or the protozoan parasite Leishmania donovani. This manuscript contains novel information on the host cell pathways targeted by Leishmania to down-modulate the expression of pro-inflammatory genes. However, a number of issues remain to be addressed to strengthen the conclusions of this study.

1. It is unclear whether the pathway described for L. donovani also works for other Leishmania species known to induce an inflammatory response, such as L. major or L. mexicana. Taking advantage of these differences among Leishmania species would strengthen the conclusions of this manuscript.

We have done experiments with two other species of Leishmania and have observed similar effect that they have on HuR downregulation upon infection of macrophage cells (Figure 5K). However following editor's suggestion we have limited our experiments using other species of *Leishmania* to explore their effect of host cell HuR protein.

Related to this, the fact that another strain of L. donovani used in this study (the Sb-resistant strain BHU569), which was shown to induce higher levels of IL-10 than the AG83 strain, also stimulates the expression of higher levels of PP2A raises the issue of mechanism. The authors refer to the Sb-S and Sb-R strains, but offer little explanations for the differential ability to stimulate IL-10 or PP2A expression. Is it related to the expression of a Leishmania gene related resistance to antimony? What is the molecular basis for antimony resistance in that strain?

The BHU569 strain used in this work is one of the clinical isolates previously characterized in terms of their sensitivity towards anti-leishmanial drug sodium stibogluconate (SSG) in macrophage culture system by Mukhopadhyay et al, in 2011. Also they had studied a series of genes related to SSG transport in the parasites itself and the resistant parasites were shown to overexpress MRPA gene higher than the sensitive one along with the higher expression of an unique terminal glycoconjugate N-acetyl-D-galactosaminy on cell membrane of resistant parasites was found. Higher IL-10 production and higher level of MDR1 expression in macrophages infected with of *Ld^R* strains comparative to the *Ld^S* strain was also reported by them (Mukhopadhyay et al, 2011).

Later the same group revealed the molecular mechanism behind higher IL-10 production and MDR1 expression in case of resistant strain infection which showed that the terminal glycoconjugate interacts with TLR2/6 which resulted in more IL-10 production via upregulation of phospho ERK level than the sensitive one (Ag83). The higher MDR1 expression was found to be IL-10 driven (Mukherjee et al, 2013).

Here our observation reveals that the BHU569 strain can produce more IL-10 upon infection via ERK1/2 pathway by higher upregulation of phospho-ERK1/2 level than the sensitive Ag83 strain. BHU569 strain also upregulated host cell MDR1 protein compared to the sensitive strain. In this work we have found that the Sb^R strain infection triggers PP2A level more than the Sb^S strain and also produces more IL-10. However, no direct relation has been found between higher IL-10 production and higher PP2A level. One such work reported previously in case of experimentally generated paromomycin resistant *Leishmania donovani*, increased level of MRPA, MDR1 and PP2A was found in the resistant parasites compared to their sensitive counterparts along with increased IL-10 production upon infection (Bhandari et al, 2014). Other groups have reported that a protein tyrosine phosphatase PTPN3 gene expression increased in cisplatin and doxorubicin resistant ovarian cancer cells (Li et al, 2016). All these previous reports suggest protein phosphatases involvement in drug resistance phenomenon.

Our observation reveals higher expression of IL-10, MDR1 along with protein phosphatase 2A upon *Ld^R* infection. Hence correlating with previous reports it can be assumed that protein phosphatase 2A might be involved in regulation of MDR1 expression and responsible for the increased pathogenicity and drug resistance phenomenon of these antimony resistant parasites. However, we have preferred not to diversify the manuscript to the extent of identification of the exact mechanism of PP2A upregulation rather its implication and mechanistic detail as how to this can be related to a differential modulation of miRNA-mediated gene repression pathway that can lead to immunomodulation to a different extent happening with the resistant parasite.

2. One major weakness of this manuscript is the absence of mechanism to explain how *Leishmania* alters the host cell processes under investigation. For instance, what triggers PP2A expression in cells exposed to *Leishmania donovani*? Is it the phagocytic process? This can be addressed by including inert particles as a phagocytosis control. The possibility that the surface glycolipid LPG is involved in this process also deserves to be investigated, since this *Leishmania* virulence factor activates TLR4. The role of LPG could be addressed using LPG mutants for *L. donovani*. Similarly, the authors provide indirect evidence suggesting that the metalloprotease GP63 cleaves HuR in infected macrophages. Given the availability of the a *L. major* GP63-deficient mutant, the authors could take the opportunity to directly assess the role of GP63 in HuR.

We thank the reviewer for pointing these out. We would like to humbly point out the major strength of this manuscript, that is to find a balancing act that PP2A and HuR expression does on miRNA-mediated repression processes to determine the inflammatory responses in mammalian macrophage cells. In this context we have utilized the *Ld* infection and LPS stimulation of macrophage where the reciprocal effect on inflammatory pathways could readily be assessed and where we found PP2A and HuR acted reversibly to determine the resultant inflammatory context of macrophage.

We have also been able to identify a unique way the parasite targets the pro-inflammatory pathways controlled by miRNAs by degrading HuR. Role of HuR in induction of inflammatory response has also been documented in this manuscript. Therefore, identification of PP2A induction mechanism will only be another additional aspect to add to the already important findings list.

PP2A expression in cells exposed to *Ld* was also an interesting question for us. We did the experiments as suggested by the reviewer and used latex beads to score the effect of phagocytosis process. Interestingly, use of latex bead to induce the phagocytosis process resulted in PP2A downregulation rather than its upregulation observed in *Ld* infection condition. However, LPG isolated from leishmania membrane, increases in PP2A level that get blocked with anti-TLR4 antibody. It suggests a TLR4-mediated signalling, controlled by LPG on parasite membrane, can increase the PP2A levels. In subsequent experiments, we have found the importance of high ERK signalling pathway prevalent in infection with drug resistant form of the parasite being linked with higher PP2A levels (See the Figure below).

Figure I. Effect of *Ld* infection on PP2A expression.

- Levels of PP2A mRNA in RAW 264.7 cells infected with drug sensitive Ag83 and two resistant strains of *Ld*.
- Effect of ERK pathway inhibitor PD98059 on PP2A mRNA expression in RAW 264.7 cells upon infection with drug sensitive Ag83 and drug resistant BHU569 strains of *Ld*.

The summary of the findings are now part of the Synopsis summary picture introduced in Figure 8 of the final version.

Regarding the importance of GP63 in cleavage of HuR in the target macrophage, it has now been addressed in detail (please see our reply to Reviewer1's point No 28). We have used liposomes reconstituted with purified GP63 to show the HuR cleavage activity of GP63 *in vivo* and in mouse model (Figure EV3 F-I and Figure 7E). However, Suggestion on the use of *L. major* GP63 deficient parasites is well taken and would be a nice suggestion for a separate work on GP63 specific role of *Leishmania* in miRNA export pathway alteration that is currently under progress in the laboratory.

We once again thank the reviewer to bring this issue to our notice to allow us to do more experiments and to make the manuscript more comprehensive and informative in all possible aspects.

3. For Fig 4A, at the bottom of page 12, the authors state that "this was accompanied by increased production of pro-inflammatory cytokines". This is a misleading statement, as the authors have measured mRNA levels by PCR. In this regard, the information related to the measurement of cytokine mRNAs is missing in the Materials and Methods section. Additionally, on Fig 4A, "Relative level IL-10 or IL-6" has to be changed to reflect the reality. The same applies to Fig 5I. In fact, actual cytokine levels should be measured.

We rectified the text and legends as appropriate to depict the changes we observed with mRNAs of cytokines. However, following the suggestion, we did measure the changes in cytokines levels by ELISA for couple of candidate cytokines to find similar effect both at protein and RNA levels for Figure 4A and 5I.

4. In Fig 4B, C, E, quantification of phagocytosis is necessary. In addition, the authors must assess parasite burden at 2h and 24 h post-phagocytosis in order to clarify whether PP2A is involved in phagocytosis per se, or controls a process associated to the microbicidal function of macrophages.

Internalized parasites from Figure 4B have been statistically represented in Figure 4C. Internalized parasites count of Figure 4E has been added in 4F. Also, the parasite count per 100 macrophage cells has been shown at 2 hours and 24hours both in presence and absence of okadaic acid in Figure 4G. The experiments described in Figure 4 E and F, where the experiments were performed with OA the inhibitor of PP2A, we have problem of parasite internalization with cells expressing wild type Ago2 but phosphorylation defective Ago2 expression allow internalization and infection. Therefore, only PP2A inhibition may not be involved in defective internalization of *Ld*. These results clarify that the PP2A may not be linked with internalization process although the internalization of parasite closely follows the expression of PP2A to have the effect downstream during the macrophage adaptation with infection.

5. Fig 7F and G, in addition to the liver, the authors should analyze what is going on in the spleen in terms of parasite burden and cytokine levels. It would also be appropriate to show the expression of HA-HuR in the organs of mice injected with the Ha-HR expression vector.

The expression of HA-HuR was found to be undetected in spleen compared to liver tissues. This is not surprising as we used the tail-vein injection of plasmid DNA as the introduction route where it is known to go and express primarily in liver only (Ghosh et al, 2013)! As expected we did not find any change in cytokine expression in spleen as well. *Ld* colonize the spleen after its infection of liver macrophage (Stanley & Engwerda, 2007) and in mouse model we could hardly found a reasonable amount of parasite in spleen after 30 days of infection (Smelt et al, 1997) . Therefore, the change in spleen parasite burden upon HA- HuR expression was not detectable. HA-HuR expression was shown in liver samples of pCIneo and HA-HuR injected animals. The western blots have been incorporated in the main figure. This is now part of Figure 7F, J and K

References

Bandyopadhyay S, Long ME, Allen LA (2014) Differential expression of microRNAs in Francisella tularensis-infected human macrophages: miR-155-dependent downregulation of MyD88 inhibits the inflammatory response. *PLoS one* **9**: e109525

Basu S, Bhattacharyya SN (2014) Insulin-like growth factor-1 prevents miR-122 production in neighbouring cells to curtail its intercellular transfer to ensure proliferation of human hepatoma cells. *Nucleic acids research* **42**: 7170-7185

Bhandari V, Sundar S, Dujardin JC, Salotra P (2014) Elucidation of cellular mechanisms involved in experimental paromomycin resistance in *Leishmania donovani*. *Antimicrobial agents and chemotherapy* **58**: 2580-2585

Brittingham A, Chen G, McGwire BS, Chang KP, Mosser DM (1999) Interaction of *Leishmania* gp63 with cellular receptors for fibronectin. *Infection and immunity* **67**: 4477-4484

Chen YL, Huang YL, Lin NY, Chen HC, Chiu WC, Chang CJ (2006) Differential regulation of ARE-mediated TNF α and IL-1 β mRNA stability by lipopolysaccharide in RAW264.7 cells. *Biochemical and biophysical research communications* **346**: 160-168

Ghosh J, Bose M, Roy S, Bhattacharyya SN (2013) Leishmania donovani targets Dicer1 to downregulate miR-122, lower serum cholesterol, and facilitate murine liver infection. *Cell host & microbe* **13**: 277-288

Lebedeva S, Jens M, Theil K, Schwanhaussner B, Selbach M, Landthaler M, Rajewsky N (2011) Transcriptome-wide analysis of regulatory interactions of the RNA-binding protein HuR. *Molecular cell* **43**: 340-352

Li S, Cao J, Zhang W, Zhang F, Ni G, Luo Q, Wang M, Tao X, Xia H (2016) Protein tyrosine phosphatase PTPN3 promotes drug resistance and stem cell-like characteristics in ovarian cancer. *Scientific reports* **6**: 36873

Lin X, Kong J, Wu Q, Yang Y, Ji P (2015) Effect of TLR4/MyD88 signaling pathway on expression of IL-1 β and TNF- α in synovial fibroblasts from temporomandibular joint exposed to lipopolysaccharide. *Mediators of inflammation* **2015**: 329405

Lu L, Zheng L, Si Y, Luo W, Dujardin G, Kwan T, Potochick NR, Thompson SR, Schneider DA, King PH (2014) Hu antigen R (HuR) is a positive regulator of the RNA-binding proteins TDP-43 and FUS/TLS: implications for amyotrophic lateral sclerosis. *The Journal of biological chemistry* **289**: 31792-31804

Morandini AC, Chaves Souza PP, Ramos-Junior ES, Souza Costa CA, Santos CF (2013) MyD88 or TRAM knockdown regulates interleukin (IL)-6, IL-8, and CXCL12 mRNA expression in human gingival and periodontal ligament fibroblasts. *Journal of periodontology* **84**: 1353-1360

Mukherjee B, Mukhopadhyay R, Bannerjee B, Chowdhury S, Mukherjee S, Naskar K, Allam US, Chakravorty D, Sundar S, Dujardin JC, Roy S (2013) Antimony-resistant but not antimony-sensitive Leishmania donovani up-regulates host IL-10 to overexpress multidrug-resistant protein 1. *Proceedings of the National Academy of Sciences of the United States of America* **110**: E575-582

Mukherjee K, Ghoshal B, Ghosh S, Chakrabarty Y, Shwetha S, Das S, Bhattacharyya SN (2016) Reversible HuR-microRNA binding controls extracellular export of miR-122 and augments stress response. *EMBO reports* **17**: 1184-1203

Mukherjee N, Corcoran DL, Nusbaum JD, Reid DW, Georgiev S, Hafner M, Ascano M, Jr., Tuschl T, Ohler U, Keene JD (2011) Integrative regulatory mapping indicates that the RNA-binding protein HuR couples pre-mRNA processing and mRNA stability. *Molecular cell* **43**: 327-339

Mukhopadhyay R, Mukherjee S, Mukherjee B, Naskar K, Mondal D, Decuypere S, Ostyn B, Prajapati VK, Sundar S, Dujardin JC, Roy S (2011) Characterisation of antimony-resistant Leishmania donovani isolates: biochemical and biophysical studies and interaction with host cells. *International journal for parasitology* **41**: 1311-1321

Neuder LE, Keener JM, Eckert RE, Trujillo JC, Jones SL (2009) Role of p38 MAPK in LPS induced pro-inflammatory cytokine and chemokine gene expression in equine leukocytes. *Veterinary immunology and immunopathology* **129**: 192-199

Rudel S, Meister G (2008) Phosphorylation of Argonaute proteins: regulating gene regulators. *Biochem J* **413**: e7-9

Saha B, Momen-Heravi F, Furi I, Kodys K, Catalano D, Gangopadhyay A, Haraszti R, Satishchandran A, Iracheta-Vellve A, Adejumo A, Shaffer SA, Szabo G (2018) Extracellular vesicles from mice with alcoholic liver disease carry a distinct protein cargo and induce macrophage activation through heat shock protein 90. *Hepatology* **67**: 1986-2000

Shi JX, Su X, Xu J, Zhang WY, Shi Y (2012) HuR post-transcriptionally regulates TNF-alpha-induced IL-6 expression in human pulmonary microvascular endothelial cells mainly via tristetraprolin. *Respiratory physiology & neurobiology* **181**: 154-161

Smelt SC, Engwerda CR, McCrossen M, Kaye PM (1997) Destruction of follicular dendritic cells during chronic visceral leishmaniasis. *J Immunol* **158**: 3813-3821

Squadrito ML, Etzrodt M, De Palma M, Pittet MJ (2013) MicroRNA-mediated control of macrophages and its implications for cancer. *Trends in immunology* **34**: 350-359

Srikantan S, Gorospe M (2011) UneCLIPsing HuR nuclear function. *Molecular cell* **43**: 319-321

Stanley AC, Engwerda CR (2007) Balancing immunity and pathology in visceral leishmaniasis. *Immunology and cell biology* **85**: 138-147

Tang B, Xiao B, Liu Z, Li N, Zhu ED, Li BS, Xie QH, Zhuang Y, Zou QM, Mao XH (2010) Identification of MyD88 as a novel target of miR-155, involved in negative regulation of Helicobacter pylori-induced inflammation. *FEBS letters* **584**: 1481-1486

Xu T, Li L, Hu HQ, Meng XM, Huang C, Zhang L, Qin J, Li J (2018) MicroRNAs in alcoholic liver disease: Recent advances and future applications. *Journal of cellular physiology* **234**: 382-394

2nd Editorial Decision

19 December 2019

Thank you for the submission of your revised manuscript to EMBO Molecular Medicine. We have now received the enclosed reports from the referees that were asked to re-assess it. As you will see the reviewers are now globally supportive and I am pleased to inform you that we will be able to accept your manuscript pending minor following editorial amendments and a response to the minor changes commented by referee 2.

Please submit your revised manuscript within three weeks. I look forward to seeing a revised form of your manuscript as soon as possible.

***** Reviewer's comments *****

Referee #2 (Remarks for Author):

The authors should incorporate their rebuttal to my previous comments 10, 12 and 13.

Referee #3 (Comments on Novelty/Model System for Author):

The authors have revised appropriately the manuscript. Some issues remain however to be clarified in the future.

2nd Revision - authors' response

24 December 2019

Referee #2 (Remarks for Author):

The authors should incorporate their rebuttal to my previous comments 10, 12 and 13.

This has been a great suggestion and we have incorporated the corresponding part of our rebuttal to Referees' comments 12,12 and 13 in respective part of the main text discussion section.

Referee #3 (Comments on Novelty/Model System for Author):

The authors have revised appropriately the manuscript. Some issues remain however to be clarified in the future.

We are happy that the reviewer has liked the modifications and additional data incorporated. Some important futuristic issues we also hope to address in our upcoming work in near future.

Corresponding Author Name: Suvendra Nath Bhattacharyya

Manuscript Number: EMM-2019-11011V1